# Continual finite-sum minimization under the Polyak-Łojasiewicz condition

## Abstract

Given functions $f_1, \ldots, f_n$ where $f_i : \mathcal{D} \mapsto \mathbb{R}$, *continual finite-sum minimization* (CFSM) (Mavrothalassitis et al., 2024) asks for an $\epsilon$-optimal sequence $\hat{x}_1, \ldots, \hat{x}_n \in \mathcal{D}$ such that

$$\sum_{j=1}^{i} f_j(\hat{x}_i)/i - \min_{x \in \mathcal{D}} \sum_{j=1}^{i} f_j(x)/i \leq \epsilon \quad \text{for all } i \in [n].$$

In this work, we develop a new CFSM framework under the Polyak-Łojasiewicz condition (PL), where each prefix-sum function $\sum_{j=1}^{i} f_j(x)/i$ satisfies the PL condition, extending the recent result of Mavrothalassitis et al. (2024) for CFSM with strongly convex functions. We present a new first-order method that under the PL condition producing an $\epsilon$-optimal sequence with overall $\mathcal{O}(n/\sqrt{\epsilon})$ first-order oracles (FOs), where an FO corresponds to the computation of a single gradient $\nabla f_j(x)$ at a given $x \in \mathcal{D}$ for some $j \in [n]$. Our method also improves upon the $\mathcal{O}(n^2 \log(1/\epsilon))$ FO complexity of state-of-the-art variance reduction methods as well as upon the $\mathcal{O}(n/\epsilon)$ FO complexity of StochasticGradientDescent. We experimentally evaluate our method in continual learning and the unlearning settings, demonstrating the potential of the CFSM framework in non-convex, deep learning problems.

## 1 Introduction

*Finite-sum minimization* (FSM) has received a lot of attention from the optimization community due to its vast applications in supervised learning Nguyen et al. (2017); Johnson & Zhang (2013); Xiao & Zhang (2014); Defazio et al. (2014); Roux et al. (2012); Allen-Zhu (2017). Given a sequence of functions $f_1, \ldots, f_n$ where $f_i : \mathcal{D} \mapsto \mathbb{R}$, FSM asks for an $\epsilon$-optimal solution $\hat{x} \in \mathcal{D}$ such that

$$\frac{1}{n} \sum_{i=1}^{n} f_i(\hat{x}) - \min_{x \in \mathcal{D}} \frac{1}{n} \sum_{i=1}^{n} f_i(x) \leq \epsilon. \tag{1}$$

A key application of FSM is the prevalent *Empirical Risk Minimization* (ERM). For example given $n$ training data points $(y_1, z_1), \ldots, (y_n, z_n)$ and a parametric model $\mathcal{M}_x(\cdot)$, ERM asks for the optimal parameters, $x^\star = \arg\min_x \sum_{i=1}^{n} \ell(\mathcal{M}_x(y_i), z_i)/n$ where $\ell(\cdot, \cdot)$ is an adequate loss function. The latter setting can be naturally captured by the FSM framework by considering $f_i(x) := \ell(\mathcal{M}_x(y_i), z_i)$.

First-order methods have long been the preferred choice for solving Problem 1. Computing a single gradient $\nabla f_j(x)$ for some $j \in [n]$ comes with a computational cost. We refer to such a computation as a *first-order oracle* (FO). The overall number of FOs that a first-order method needs to solve Problem 1 defines the FO complexity of the method. Since $n$ can be of the order of millions in modern Machine Learning applications, a crucial desideratum in FSM is the design of first-order methods that scale efficiently with $n$ and $1/\epsilon$. *Variance-reduction methods* (VR) were able to fulfill the latter goal by achieving optimal FO complexity for FSM under various assumptions on the functions (strongly convex, convex, non-convex, Polyak-Łojasiewicz) Nguyen et al. (2017); Johnson & Zhang (2013); Xiao & Zhang (2014); Defazio et al. (2014); Roux et al. (2012); Allen-Zhu (2017); Karimi et al. (2016); Lei et al. (2017); Zhou et al. (2018); Li et al. (2021); Zhang et al. (2016).

*Continual Learning and Finite-Sum Minimization:* Continual Learning refers to the process of continuously updating a model as new data become available Castro et al. (2018); Rosenfeld & Tsotsos

(2018); Hersche et al. (2022). This approach enables machine learning systems to adapt to changing environments. Incorporating new data without completely retraining the model is highly challenging because it can greatly reduce the model's effectiveness on past data (*catastrophic forgetting* Castro et al. (2018); Goodfellow et al. (2014); Kirkpatrick et al. (2017); McCloskey & Cohen (1989)). Similar challenges of *time-evolving data sets* arise in the more recent context of *unlearning* where goal is to remove data previously present in the training set Sekhari et al. (2021); Guo et al. (2020).

We can simply resolve the FSM (Problem 1) once a new data point is added as datasets evolve. Unfortunately this approach is wasteful due to the computational costs that it incurs. In their recent work, Mavrothalassitis et al. Mavrothalassitis et al. (2024) introduced a twist of FSM, called *Continual Finite-Sum Minimization* (CFSM) in order to formally examine the new challenge.

**Definition 1.** *Given a sequence of functions $f_1, \ldots, f_n$ where $f_i : \mathcal{D} \mapsto \mathbb{R}$, Continual Finite-Sum Minimization asks for a sequence of $\epsilon$-approximate solutions $\hat{x}_1, \ldots, \hat{x}_n \in \mathcal{D}$ such that*

$$\frac{1}{i} \sum_{j=1}^{i} f_j(\hat{x}_i) \ \leq \ \min_{x \in \mathcal{D}} \frac{1}{i} \sum_{j=1}^{i} f_j(x) \ + \ \epsilon \quad \textit{for each stage } i \in [n].$$

To understand why CFSM captures the continual learning setting, let the training set be initially composed by the first $i$ data points, $(y_1, z_1), \ldots, (y_i, z_i)$. In this case, ERM asks for $x_i^\star := \arg\min_x \sum_{j=1}^{i} \ell(\mathcal{M}_x(y_j), z_j)/i$. Assume that after computing an $\epsilon$-optimal point $\hat{x}_i$, a new data point $(y_{i+1}, z_{i+1})$ is revealed to the model. Now the model's parameters need to be updated to $x_{i+1}^\star := \arg\min_x \sum_{j=1}^{i+1} \ell(\mathcal{M}_x(y_j), z_j)/(i+1)$.

**Example 1.** *(**Unlearning**) An interesting application of CFSM is unlearning where the goal is to remove data points used in the training of the model Sekhari et al. (2021); Guo et al. (2020). For example, assume that we have trained a model $\mathcal{M}_x(\cdot)$ with respect to some initial data $S$, meaning that*

$$x_{init} := \arg\min_{x \in \mathbb{R}^d} \left[ \sum_{i \in S} \ell(\mathcal{M}_x(y_i), z_i) \right].$$

*Now, let us say that we want to remove a small subset of data points $F \subseteq S$, meaning that we would like to update the model's parameters to*

$$x_{updated} := \arg\min_{x \in \mathbb{R}^d} \left[ \sum_{i \in S/F} \ell(\mathcal{M}_x(y_i), z_i) := \sum_{i \in S} \ell(\mathcal{M}_x(y_i), z_i) - \sum_{i \in F} \ell(\mathcal{M}_x(y_i), z_i) \right].$$

*Retraining the model from scratch to compute $x_{updated}$ is not computationally viable. CFSM nicely captures this setting by considering the arrival of the function $-\sum_{j \in F} \ell(\mathcal{M}_x(y_j), z_j)$.*

Mavrothalassitis et al. Mavrothalassitis et al. (2024) introduces an approximately optimal method with respect to the overall FO complexity (across all $n$ stages) for CFSM under the assumption that each prefix-sum function $g_i(x) := \sum_{j=1}^{i} f_j(x)/i$ is $\mu$-strongly convex. They provide a first-order method for CFSM with overall $\tilde{\mathcal{O}}(n/\epsilon^{1/3})$ FOs. The latter complexity improves upon the $\mathcal{O}(n^2 \log(1/\epsilon))$ of state-of-the-art VR methods for the strongly convex case Nguyen et al. (2017); Johnson & Zhang (2013); Xiao & Zhang (2014); Defazio et al. (2014); Roux et al. (2012); Allen-Zhu (2017) and the $\mathcal{O}(n/\epsilon)$ FO complexity of StochasticGradientDescent.

**Our Contribution and results:** In this work, we build upon the work of Mavrothalisitis et al. Mavrothalassitis et al. (2024) by providing an efficient first-order method for CFSM in the *non-convex* regime. More precisely, we consider CFSM under the assumption that each prefix-sum function $g_i(x) := \sum_{j=1}^{i} f_j(x)/i$ satisfies the Polyak-Łojasiewicz (PL) condition. The PL condition is a weaker assumption than strong convexity that has received huge attention in recent years due to the fact that it provides structured non-convexity for optimization settings related to the training of Deep Neural Networks Liu et al. (2022); Song et al. (2021) (see also Remark 2).

Finite-sum minimization (Problem 1) under the PL condition has been previously considered in the variance-reduction literature Karimi et al. (2016); Lei et al. (2017); Li et al. (2021); Zhang et al. (2016). However as in the strongly convex case using such a method in a black-box manner (at each stage $i \in [n]$) leads to an overall $\mathcal{O}(n^2 \log(1/\epsilon))$ FO complexity. Despite that

the dependence $\mathcal{O}\left(\log(1/\epsilon)\right)$ seems appealing, the $\mathcal{O}(n^2)$ is extremely computationally heavy even in relatively small cases where $n$ is of the order of thousands. At the same time, using StochasticGradientDescent at each stage $i \in [n]$ would require $\mathcal{O}(n/\epsilon)$ FO complexity that is still impractical due to the coupled dependence of $\mathcal{O}(n)$ and $\mathcal{O}(1/\epsilon)$.

The main contribution of this work is providing a first-order method, called *CSVRG-PL*, achieving $\tilde{\mathcal{O}}(n/\sqrt{\epsilon})$ FO complexity for CSFM under the Polyak-Łojasiewicz condition.

**Main Result.** *There exists a first-order method, called CSVRG-PL (Algorithm 1), for Continual Finite-Sum Minimization under Polyak-Łojasiewicz condition that achieves $\tilde{\mathcal{O}}(n/\sqrt{\epsilon})$ FO complexity.*

Different methods for CFSM might be more efficient than others depending on the required accuracy $\epsilon > 0$. For accuracy $\epsilon = \Theta(1/n)$, CSVRG-PL (our method) requires only $\mathcal{O}(n^{3/2})$-FOs while both the variance reduction methods Karimi et al. (2016); Lei et al. (2017); Li et al. (2021); Zhang et al. (2016) and StochasticGradientDescent require $\mathcal{O}(n^2)$ FOs. As also commented in Mavrothalassitis et al. (2024), the accuracy regime $\epsilon = \Theta(1/n)$ is of particular interest since the statistical error of empirical risk minimization is $\Theta(1/n)$ Shalev-Shwartz et al. (2010) and thus requiring accuracy $\epsilon > 0$ smaller than $\Theta(1/n)$ is redundant in the context of empirical risk minimization (see also Bottou & Bousquet (2007)).

**Remark 1.** *Our method CSVRG-PL (Algorithm 1) achieves almost optimal FO complexity. Mavrothalassitis et al. Mavrothalassitis et al. (2024) showed that even in the strongly convex case there is no first-order method achieving $\mathcal{O}(n/\epsilon^{1/4})$-FOs meaning that our method is only $\mathcal{O}(1/\epsilon^{1/4})$ far from being optimal. At the same time, Mavrothalassitis et al. Mavrothalassitis et al. (2024) also established that in the strongly convex case there is no first-order method with $o(n^2 \log(1/\epsilon))$ FO complexity. The latter implies that the $\log(1/\epsilon)$ FO complexity cannot be achieved without a quadratic dependence on $n$ on the total FO oracles.*

**Remark 2.** *Neural Networks and Polyak-Łojasiewicz Condition: Analyzing the convergence of methods and optimization algorithms for neural networks, has been a challenging endeavor, despite their empirical success LeCun et al. (1998a; 2015); Zhang et al. (2021), due to the highly non-convex landscape. A widely used approach, for studying optimization of neural networks, has been the Neural Tangent Kernel (NTK) Jacot et al. (2018). There are several works, that are based on this approach Awasthi et al. (2021); Su & Yang (2019); Zou & Gu (2019), however, such analysis requires heavy overparametrization, which is extremely costly for deep neural networks. Another approach is based on structural hypothesis of the loss landscape, of neural networks, using the local Polyak-Łojasiewicz Condition Song et al. (2021); Nguyen (2021); Ling et al. (2023). This approach has the benefit of decoupling the neural network dynamics from the optimization analysis.*

Table 1: Convergence Results for continual finite sum minimization(CFSM)

| Method | Number of FOs | Assumption |
|---|---|---|
| StochasticGradientDescent | $\mathcal{O}(n/\epsilon)$ | strongly convex |
| Katyusha(Allen-Zhu, 2017) | $\mathcal{O}\left(n^2 \log(1/\epsilon)\right)$ | strongly convex |
| CSVRG(Mavrothalassitis et al., 2024) | $\tilde{\mathcal{O}}\left(n/\epsilon^{1/3}\right)$ | strongly convex |
| StochasticGradientDescentLei et al. (2019) | $\mathcal{O}(n/\epsilon)$ | Polyak-Łojasiewicz |
| SpiderBoost($W$ $an$ $g$ $et$ $al.$, 2019) | $\mathcal{O}\left(n^2 \log(1/\epsilon)\right)$ | Polyak-Łojasiewicz |
| CSVRG-PL (our method) | $\tilde{\mathcal{O}}\left(n/\sqrt{\epsilon}\right)$ | Polyak-Łojasiewicz |

Up next, we summarize the contributions our work:

**1**. We provide the $\mathrm{CSVRG-PL}$ algorithm, a method with provable guarantees (both in terms of convergence and FO complexity) for CFSM under the Polyak-Łojasiewicz condition. The latter provides a solid theoretical foundation of the performance of our method in settings involving Deep Neural Networks Mavrothalassitis et al. (2024); Nguyen (2021); Ling et al. (2023).

**2**. We provide extensive experimental evaluations in *continual learning* Castro et al. (2018); Rosenfeld & Tsotsos (2018); Hersche et al. (2022) and *unlearning settings*Sekhari et al. (2021); Guo et al. (2020) involving baseline datasets (MNIST, FashionMNIST, CIFAR-10, CIFAR-100) in ResNet18. Our experimental evaluations reveal a significant improvement in the performance of our method

compared to $\mathrm{StochasticGradientDescent}$. Furthermore, our evaluations highlight the potential of both our method and the CFSM framework in handling settings with evolving datasets. We note that our experimental evaluations substantially extend beyond those of Mavrothalassitis et al. (2024), which only considers linear regression on the LIBSVM dataset.

**3**. Our CFSM-PL method shares the same algorithmic architecture with the CFSM method in Mavrothalassitis et al. (2024), with one key difference. The estimator that we use in Step 5 of Algorithm2 differs from the respective estimator in Mavrothalassitis et al. (2024) (see the respective FUM method). This difference may seem subtle but is of great importance for providing formal convergence guarantees under the Polyak-Łojasiewicz condition. Specifically, the current estimator allows us to provide *last-iterate guarantees* during each call of Algorithm 2. In contrast, the estimator used by Mavrothalassitis et al. Mavrothalassitis et al. (2024) only provide *time-average guarantees*. While this limitation is not critical in the strongly convex regime, it poses a significant problem in the non-convex regime considered here. Additionally, the estimator in Mavrothalassitis et al. (2024) is significantly more complex than ours and, most importantly, requires three FOs. In contrast, the current estimator requires only two FOs, making our method $1.5$ times faster for the same number of inner iterations during each stage.

## 2 PRELIMINARIES AND RESULTS

In this section we introduce some basic definitions and notation. We denote with $\mathrm{Unif}(1, \ldots, n)$ the uniform distribution over $\{1, \ldots, n\}$ and $[n] := \{1, \ldots, n\}$.

**Definition 2.** *A differentiable function $f : \mathbb{R}^d \mapsto \mathbb{R}$ is said to be L-smooth if and only if*

$$\|\nabla f(x) - \nabla f(y)\| \leq L \cdot \|x - y\| \text{ for all } x, y \in \mathbb{R}^d$$

**Definition 3** (Polyak-Łojasiewicz condition). *A differentiable function $f : \mathbb{R}^d \mapsto \mathbb{R}$ is said to satisfy the Polyak-Łojasiewicz (PL) condition, if for all $x \in \mathbb{R}^d$, it holds that:*

$$f(x) - f^\star \leq \frac{1}{2\mu} \|\nabla f(x)\|^2$$

*where $f^\star$ is the minimum of $f(x)$, i.e. $f^\star = f(x^\star)$ where $x^\star = \arg\min_{x \in \mathbb{R}^d} f(x)$.*

The PL condition does not enforce convexity (e.g. $f(x) = x^2 + 3\sin^2(x)$ is PL but non-convex), recent works have shown that local Polyak-Łojasiewicz can be used to characterize the loss landscape of neural networks Song et al. (2021); Nguyen (2021); Ling et al. (2023), rendering it a reasonable assumption for modern optimization analysis in settings involving deep neural nets. The PL-condition also implies the Quadratic Growth property Karimi et al. (2016) stated up next.

**Definition 4** (Quadratic Growth). *A function $f : \mathbb{R}^d \mapsto \mathbb{R}$ satisfies the Quadratic Growth property, if for all $x \in \mathbb{R}^d$, we have that:*

$$f(x) - f^\star \geq \frac{\mu}{2} \|x - x_p\|^2$$

*where $x_p$ is the projection of $x$ on the set of optimal solutions $\mathcal{X}^\star := \{x \in \mathbb{R}^d : f(x) := f^\star\}$.*

To simplify notation, we denote with $g_i(x)$ the prefix-function at stage $i \in [n]$,

$$g_i(x) := \sum_{j=1}^{i} f_j(x)/i.$$

We also denote with $g_i^\star = \min_{x \in \mathbb{R}^d} g_i(x)$ the minimum of $g_i(x)$ and with $\mathcal{X}_i^\star$ the set of optimal solutions $\mathcal{X}_i^\star = \{x \in \mathbb{R}^d \mid g_i(x) = g_i^\star\}$. Finally we will make the following two assumptions for the prefix-sum functions $g_i(x)$.

**Assumption 1.** *For any stages $i > j$, $\left\|\nabla f_i\left(x_j^\star\right)\right\| \leq L \cdot D$, for some constant $D > 0$.*

This assumption is common in stochastic optimization Stich et al. (2018); Nemirovski et al. (2009).

**Assumption 2.** *(Diminishing returns) There exists a constant $K > 0$ such that for any stages $i > j$,*

$$\max_{x_i^\star \in \mathcal{X}_i^\star, x_j^\star \in \mathcal{X}_j^\star} \left\|x_i^\star - x_j^\star\right\|^2 \leq \frac{i-j}{i} K.$$

Assumption 2 requires that any two optimal solutions $x_i^\star \in \mathcal{X}_i^\star$ and $x_j^\star \in \mathcal{X}_j^\star$ scale with $\mathcal{O}\left(\frac{i-j}{i}\right)$.

**Remark 3.** *Assumption 2 mathematically encodes diminishing returns, one of several ways we can use to capture this phenomenon. Though submodularity is a more general framework, we have chosen the simpler form presented here as it makes the convergence analysis transparent. Our assumption ensures that each function's contribution to the overall loss remains consistent, varying only by constant factors, and diminishes as the model progresses through successive stages. In the generalization literature, a related concept is that of uniform stability, as discussed in Feldman & Vondrak (2018); Bassily et al. (2020). This assumption posits that altering a single data point does not substantially affect the loss, a crucial tool when establishing generalization bounds.*

In Appendix A, we explicitly show a stronger inequality to verify the assumption under a couple general settings, such as, when the functions have a unique minimizer, or the set of minimizers is shrinking. We now present the formal statement of our result.

**Theorem 1.** *There exists a first-order method, called CSVRG-PL (Algorithm 1) such that for any $f_1, \ldots, f_n$ satisfying the PL-condition and Assumptions 1&2, outputs $\hat{x}_1, \ldots, \hat{x}_n \in \mathbb{R}^d$ such that*

$$\mathbb{E}\left[g_i(\hat{x}_i)\right] - g_i^\star \leq \epsilon \quad \text{for each stage } i \in [n]$$

*with overall $\tilde{\mathcal{O}}(n/\sqrt{\epsilon})$ first-order oracles.*

## 3 OUR METHOD AND CONVERGENCE RESULTS

In this section we present our first-order method called *Continual Stochastic Variance Reduction - Polyak-Łojasiewicz* $(\mathrm{CSVRG} - \mathrm{PL})$ that is able to achieve the guarantees of Theorem 1. $\mathrm{CSVRG} - \mathrm{PL}$ is formally described in Algorithm 1 and is composed by two main components (Algorithm 1 and Algorithm 2) that we subsequently explain.

First notice that that the output $\hat{x}_i \in \mathbb{R}^d$ at each stage $i \in [n]$ is provided by Algorithm 2, which can be viewed as a version of stochastic gradient descent that at each iteration $t \in [T_i]$ follows the direction
$$\nabla_i^t \leftarrow \nabla f_{u_t}(x_i^t) - \nabla f_{u_t}(\hat{x}_{\text{prev}}) + \tilde{\nabla}_i \quad \text{(Step 5 of Algorithm 2)}.$$

The only purpose of Algorithm 1 is to update the estimator $\tilde{\nabla}_i$ at each stage $i \in [n]$. The latter is then given as input to Algorithm 2 (Step 13 of Algorithm 1). Notice that the way $\tilde{\nabla}_i$ is calculated differs from stage to stage. More precisely,

$$\tilde{\nabla}_i = \begin{cases} \frac{1}{i} \sum_{j=1}^i \nabla f_j(\hat{x}_{i-1}) & i - \text{prev} \geq \alpha \cdot i \\ \left(1 - \frac{1}{i}\right) \tilde{\nabla}_{i-1} + \frac{1}{i} \nabla f_i(\hat{x}_{\text{prev}}) & i - \text{prev} < \alpha \cdot i \end{cases}$$

In case $i - \text{prev} \geq \alpha \cdot i$, Algorithm 1 requires $i$ FOs to calculate $\tilde{\nabla}_i$ while in case $i - \text{prev} < \alpha \cdot i$ only 1 additional FO is required, specifically $\nabla f_i(\hat{x}_{\text{prev}})$. For this reason by the construction of Algorithm 1 in *almost all stages* 1 FO is used for the calculation of $\tilde{\nabla}_i$. The latter is ensured by Steps (14)-(16) of Algorithm 1 guaranteeing that $(i - \text{prev} \geq \alpha \cdot i)$ is satisfied more and more sparsely over the sequence. Finally we note that $\tilde{\nabla}_i = \nabla g_i(\hat{x}_{\text{prev}})$ always at Step 13 of Algorithm 1, as shown in Lemma 6.

Up next we explain the main ideas used in the analysis and design of Algorithm 1. The cornerstone of our analysis comes from the fact that the estimator $\nabla_i^t$ in Step 5 of Algorithm 2 is always an unbiased estimator of the true gradient $\nabla g_i(x_i^t)$. The latter is formally stated and established in Lemma 1.

**Lemma 1** (Unbiased). *Let $\nabla_i^t$ be the gradient estimator as defined in Step 5 of Algorithm 2. Then for all $t \in [T_i]$ and for all $i \in [n]$, it holds that: $\mathbb{E}\left[\nabla_i^t\right] = \nabla g_i(x_i^t)$.*

The proof of Lemma 1 lies in Appendix B. The proof heavily relies on the fact that $\tilde{\nabla}_i = \nabla g_i(\hat{x}_{\text{prev}})$ is guaranteed by Algorithm 1. The fact that $\mathbb{E}\left[\nabla_i^t\right] = \nabla g_i(x_i^t)$ is obviously a very crucial property since it guarantees that on expectation Algorithm 2 decreases the function $g_i(x) := \sum_{j=1}^i f_j(x)/i$.

To this end one might wonder why Algorithm 2 needs such an intricate gradient estimator, for example the straightforward estimator $\nabla_i^t := \nabla f_{u_t}(x_i^t)$ where $u_t \sim \text{Unif}(1, \ldots, n)$ still satisfies the unbiasness property. The problem with the latter estimator is that it admits very high variance,

---

**Algorithm 1** CSVRG − PL

---

1: $\hat{x}_0 \in \mathcal{D}$, prev $\leftarrow 0$, $update \leftarrow false$
2: $\hat{x}_1 \leftarrow \text{GradientDescent}(\hat{x}_0)$
3: $\tilde{\nabla}_1 \leftarrow \nabla f_1(\hat{x}_1)$
4: **for** each stage $i = 2, \ldots, n$ **do**
5:     **if** $i - \text{prev} \geq \alpha \cdot i$ **then**
6:         $\tilde{\nabla}_i \leftarrow \frac{1}{i} \sum_{j=1}^{i} \nabla f_j(\hat{x}_{i-1})$                      ▷ Full gradient $i$ FOs
7:         prev $\leftarrow i - 1$
8:         $update \leftarrow true$
9:     **else**
10:         $\tilde{\nabla}_i \leftarrow \left(1 - \frac{1}{i}\right) \tilde{\nabla}_{i-1} + \frac{1}{i} \nabla f_i(\hat{x}_{\text{prev}})$     ▷ Update full gradient with 1 FO
11:     **end if**
12:     $T_i \leftarrow \mathcal{O}\left(\frac{L^2 G}{\mu^{5/2} i \sqrt{\epsilon}} + \frac{L^2 G^2 \alpha^2}{\mu \epsilon} + \frac{L^2}{\mu^2}\right)$     ▷ Number of iterations at stage $i \in [n]$
13:     $\hat{x}_i \leftarrow \text{FUM} - \text{PL}(\text{prev}, \tilde{\nabla}_i, T_i)$     ▷ Output for stage $i$ by Algorithm 2
14:     **if** $update$ **then**
15:         $\tilde{\nabla}_i \leftarrow \frac{1}{i} \sum_{j=1}^{i} \nabla f_j(\hat{x}_i)$                    ▷ Full gradient $i$ FOs
16:         prev $\leftarrow i$
17:         $update \leftarrow false$
18:     **end if**
19: **end for**

---

**Algorithm 2** **F**requent **U**pdate **M**ethod-PL (FUM − PL)

---

1: $c \leftarrow \frac{8L^2}{\mu^2}$
2: $x_i^0 \leftarrow \hat{x}_{i-1}$                     ▷ Initialization at previous solution
3: **for** each round $t := 1, \ldots, T_i$ **do**
4:     Select $u_t \sim \text{Unif}(1, \ldots, i)$
5:     $\nabla_i^t \leftarrow \nabla f_{u_t}(x_i^t) - \nabla f_{u_t}(\hat{x}_{\text{prev}}) + \tilde{\nabla}_i$            ▷ 2 FOs
6:     $\gamma_t \leftarrow 2/(\mu(t + c))$
7:     $x_i^{t+1} \leftarrow x_i^t - \gamma_t \nabla_i^t$                  ▷ Update $x_i^t \in \mathcal{D}$
8: **end for**
9: **Output:**  $\hat{x}_i \leftarrow x_{T_i}$                ▷ Output at stage $i \in [n]$

---

on the other hand the estimator $\nabla_i^t \leftarrow \nabla f_{u_t}(x_i^t) - \nabla f_{u_t}(\hat{x}_{\text{prev}}) + \tilde{\nabla}_i$, which was initially introduced in Johnson & Zhang (2013), admits way smaller variance which in turn translates to significantly faster rates. The latter is established in Lemma 2 that we present up next. The proof of Lemma 2 lies in Appendix C.

**Lemma 2** (Variance). *For the gradient estimator used in line* 5 *of Algorithm* 2 *we have:*

$$
\begin{aligned}
\mathbb{E}\left[\left\|\nabla_i^t - \nabla g_i(x_i^t)\right\|^2\right] &\leq 4\frac{L^2}{\mu} \mathbb{E}\left[g_i(x_i^t) - g_i(x_i^\star)\right] \\
&+ 4L^2 \left(\frac{2}{\mu} \mathbb{E}\left[(g_{prev}(\hat{x}_{prev}) - g_{prev}(x_{prev}^\star))\right] + \frac{i - prev}{i} K\right)
\end{aligned}
$$

*where $K$ is the constant of Assumption 2.*

From Lemma 2, one can notice that the variance of the estimator is upperbound by three terms. The first one $\mathbb{E}\left[g_i(x_i^t) - g_i(x_i^\star)\right]$ depends on the current iterations suboptimality, therefore as we approach an optimal point this term vanishes and this allows us to incorporate it seamlessly into the analysis. The other two terms $\mathcal{O}(\mathbb{E}\left[g_i(x_i^t) - g_i(x_i^\star)\right])$ and $\mathcal{O}(\frac{i-j}{i})$ are independent of the iteration

$t \in [T_i]$. Therefore we cannot treat them between individual iterations and instead they appear in the upper bound of the suboptimality of Algorithm 2.

Specifically, using Lemma 2 we can then upper bound the suboptimality of the output $\hat{x}_i$ of Algorithm 2. The latter is formally stated and proven in Lemma 3 the proof of which lies in Appendix F.

**Lemma 3** (Convergence). *Under the update rule of line 5 of Algorithm 2, when using the step size* $\gamma_t = 2/(\mu(t+c))$, *with* $c = 4L(\mu^2 + L^2)/\mu^3$, *we have that:*

$$
\begin{aligned}
\mathbb{E}\left[g_i(x_i^{T+1}) - g_i(x_i^\star)\right] &\leq (c-1)(c-2)\frac{\mathbb{E}\left[g_i(\hat{x}_{i-1})\right] - g_i(x_i^\star)}{T^2} + 2L^3 K \frac{i - prev}{iT} \\
&+ \frac{16L^3\left(\mathbb{E}\left[(g_{prev}(\hat{x}_{prev}) - g_{prev}(x_{prev}^\star))\right]\right)}{\mu^3 T}
\end{aligned}
$$

The suboptimality achieved by Algorithm 2 depends on three different terms, respectively $\mathcal{O}\left(\mathbb{E}\left[g_i(\hat{x}_{i-1})\right] - g_i(x_i^\star)/T^2\right)$, $\mathcal{O}\left((i - \text{prev})/(iT)\right)$ and $\mathcal{O}\left(\left(\mathbb{E}\left[(g_{\text{prev}}(\hat{x}_{\text{prev}})\right] - g_{\text{prev}}(x_{\text{prev}}^\star)\right)/T\right)$. In order to analyze their convergence, we will have to rely on the expected suboptimality of the solutions produced by Algorithm 2 in the previous stages. We can inductively assume that we have calculated a solution $\hat{x}_j$, for all $j < i$, such that: $\mathbb{E}\left[g_j(\hat{x}_j)\right] - g_j(x_j^\star) \leq \epsilon$. This inductive hypothesis, allows us to directly bound the third term of the aforementioned bound directly.

The first term $\left(\mathbb{E}\left[g_i(\hat{x}_{i-1})\right] - g_i(x_i^\star)\right)/T^2$ depends on the suboptimality of the point $\hat{x}_{i-1}$. Bounding this term can be done using our inductive hypothesis and Lemma 4 the proof of which can be found in Appendix E.

**Lemma 4.** *Let a sequence of functions* $f_1, f_2, \ldots, f_i$ *such that Assumption 2 is satisfied and the functions* $g_i, g_j$ *satisfy the quadratic growth property. Then,*

$$
g_i(\hat{x}_j) - g_i^\star \leq \frac{i-j}{i}\frac{L \cdot K}{2} + C
$$

*where* $C = \frac{i-j}{i}L \cdot D \cdot \sqrt{\frac{2}{\mu}\left(g_j(\hat{x}_j) - g_j^\star\right)} + \frac{L}{\mu}\left(g_j(\hat{x}_j) - g_j^\star\right)$ *and* $K$ *is the constant in Assumption 2.*

Applying Lemma 4 for $j = i - 1$ we get that $\mathbb{E}\left[g_i(\hat{x}_{i-1})\right] - g_i(x_i^\star) \leq L \cdot K/(2 \cdot i) + \mathbb{E}\left[C\right]$ and our inductive hypothesis guarantees that $\mathbb{E}\left[C\right] \leq \mathcal{O}\left(\sqrt{\epsilon}\right)$.

Finally, the second term $\mathcal{O}((i - \text{prev})/i)$ quantifies the distance between stage $i$ and prev, i.e. the most recent stage at which Algorithm 1 computation got into Step $(5) - (6)$. Notice that by the construction of Algorithm 1, $i - \text{prev} < \alpha \cdot i$ meaning that the term $\mathcal{O}((i - \text{prev})/i) \leq \alpha$. To this end we remark that the parameter $\alpha$ controls the frequency according to which Algorithm 1 gets into Steps $(5) - (8)$. Lemma 3 reveals that the higher the frequency (smaller $\alpha$) the better the rate of Algorithm 2 during stage $i$. However we remark that once Algorithm 1 gets into Step $(5) - (8)$ more FOs are needed, thus a proper selection of $\alpha$ is required in order to provide the desirable results.

Up next we present Theorem 2 that describes the overall FO complexity of our method. The proof of Theorem 2 lies on Appendix G and is based on the arguments that we illustrated above.

**Theorem 2.** *Let Algorithm 1 with parameters* $\alpha = \sqrt{\epsilon/(10L^3 K)}$, $\gamma_t = 2/(\mu(t+c))$ *and,*

$$
T_i = \max\left(\sqrt{\frac{5LK}{2i\epsilon}}, c \cdot \sqrt{\frac{5L \cdot D}{i\sqrt{\epsilon}}}\sqrt{\frac{2}{\mu}}, \frac{80L^3}{\mu^3}, \sqrt{\frac{10L^3 K}{\epsilon}}\right)
$$

*where* $c = 4L(\mu^2 + L^2)/\mu^3$. *Then for each stage* $i \in [n]$ *we have that*

$$
\mathbb{E}\left[g_i(\hat{x}_i)\right] - g_i^\star \leq \epsilon
$$

*where* $\hat{x}_i$ *is the output of Algorithm 2, for stage* $i$.

We complete the section via providing the overall number of FOs that Algorithm 1 requires under the selection of parameters described in Theorem 2. Counting the overall number of FOs that Algorithm 1 requires is tedious but relative straightfoward and is based on Corollary 1, the proof of which can be found in Appendix H

**Corollary 1.** *Over a sequence of $n$ stages, Algorithm 1 requires $2\sum_{i=1}^{n} T_i + 2n\lceil \log n/\alpha \rceil$ FOs.*

Up next we present a proof sketch. The term $2\sum_{i=1}^{n} T_i$ comes form the the fact that at each iteration $t \in [T_i]$ of Algorithm 2, 2 FOs are calculated, $\nabla \tilde{f}_{u_t}(x_i^t)$ and $\nabla f_{u_t}(x_{\text{prev}})$. At the same time the term $2n\lceil \log n/\alpha \rceil$ accounts for the additional FOs that Algorithm 1 uses in order to maintain $\tilde{\nabla}_i$. The fact that Algorithm 1 sporadically computes needs $i$ FOs ($i - \text{prev} \le \alpha \cdot i$) and in most cases only needs 1 additional FO is depicted in $2n\lceil \log n/\alpha \rceil$.

Using the parameters $\alpha$ and $T_i$ described in Theorem 2 we obtain the overall FO complexity of Algorithm 1. The latter is formally stated and proven in Theorem 3, the proof of which lies in Appendix H.

**Theorem 3.** *Let Algorithm 1 with parameters $\alpha = \sqrt{\epsilon/(10L^3K)}$, $\gamma_t = 2/(\mu(t+c))$ and,*

$$T_i = \max\left( \sqrt{\frac{5LK}{2i\epsilon}}, c \cdot \sqrt{\frac{5L \cdot D}{i\sqrt{\epsilon}}} \sqrt{\frac{2}{\mu}}, \frac{80L^3}{\mu^3}, \sqrt{\frac{10L^3K}{\epsilon}} \right)$$

*where $c = 4L(\mu^2 + L^2)/\mu^3$. Then the overall FO complexity of Algorithm 1 (across all $n$ stages) equals $\mathcal{O}\left( n \log n \sqrt{1/\epsilon} \right)$.*

## 4 EXPERIMENTS

We validate our algorithm and setup by training the popular PreActResNet18 architecture (He et al., 2016) in the MNIST (LeCun et al., 1998b), FashionMNIST (Xiao et al., 2017) and CIFAR10/100 (Krizhevsky & Hinton, 2009) datasets. As a baseline, we compare against SGD. Please note that as we mention in the introduction, traditional variance reduction methods such as SVRG require $\Omega(n^2)$ FOs, making them impractical to run, this is clearly demonstrated in Table 2 of Mavrothalassitis et al. (2024).

Below, we use a batch size $b = 100$ samples and $\alpha = 0.01$ (see Algorithm 1) and report the average performance over three random seeds. We also included more extensive experiments in Appendix I.

### 4.1 LEARNING RATE AND SCHEDULER SELECTION

We consider two learning rate schedulers:

1. *constant*: the learning rate remains constant at each stage and FUM update.
2. *linear*: we use a base learning rate $\gamma$ and inside every FUM loop (Algorithm 2), we decrease linearly from $\gamma$ to $\gamma/10$.

We perform a grid search to select the learning rate for $\text{CSVRG} - \text{PL}$ and SGD in the constant and linear schedules, we leave out a subset of $5\%$ of the training data as a validation set. We run every method and learning rate value for 3 random seeds and report the average and the final accuracy on the validation set. We select the optimal learning rate, according to Table 4, presented in detail in Appendix I.1. We also find that the linear scheduler improves the performance for both methods.

### 4.2 CONTINUAL LEARNING

In this section we evaluate our method in a continual learning setting Castro et al. (2018); Rosenfeld & Tsotsos (2018); Hersche et al. (2022). We split the dataset according to the labels. As a reference, in CIFAR10 we create ten subsets of the data, each one corresponding to a label, from "0" to "9". After splitting the data, we start training the model. Initially the training set of the model is empty and we add to it one batch of samples at each stage, introducing first all the data of one label, before moving on to the next one. Going back to our example on CIFAR10, this procedure corresponds to starting from label "0" and adding the samples with label "0" for stages 0 to 49, then, we start introducing data with label "1", for stages 50 to 99 and so on.

We measure the performance of the model at each stage with respect to the test accuracy of the labels that have been currently added in the training set of the model. We avoid doing extensive

computation at each stage, since we don't want to retrain the model from scratch on the new dataset, but instead make small adaptations to it. Up next, we compare our method with SGD in this setting.

Table 2: Test accuracy at the last stage, with only 10 iterations per stage.

| Dataset | MNIST | FashionMNIST | CIFAR10 | CIFAR100 |
|---|---|---|---|---|
| SGD | $98.34_{\pm(0.09)}$ | $91.29_{\pm(0.11)}$ | $80.24_{\pm(0.16)}$ | $54.33_{\pm(0.57)}$ |
| CSVRG − PL | $\mathbf{98.69}_{\pm(0.17)}$ | $\mathbf{91.52}_{\pm(0.51)}$ | $\mathbf{85.54}_{\pm(0.44)}$ | $\mathbf{61.12}_{\pm(1.00)}$ |

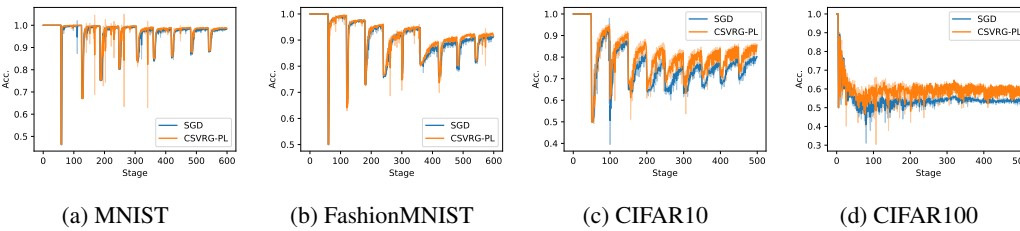

| (a) MNIST | (b) FashionMNIST | (c) CIFAR10 | (d) CIFAR100 |
|---|---|---|---|

Figure 1: We report the test accuracy in the visited classes. We observe CSVRG − PL recovers accuracy much faster than SGD when a new class is introduced and attains a much higher final accuracy.

In Fig. 1 we can see that our method consistently outperforms SGD, on test accuracy. At stages where a new class is introduced the performance of the model drops regardless of the training algorithm used. The latter is totally expected since when the first few batches of a new class are introduced the model has seen very few data of the new class. For example, for CIFAR10 these stages are $50, 100, \ldots, 450$. After these drops, we can see that CSVRG − PL recovers faster than SGD and adapts its predictions to account for the new class, leading to higher accuracy throughout the experiments (see Appendix I.2). It is worth noting that on FashionMNIST and especially MNIST the performance of the algorithms is very close, due to the simplicity of the dataset, while on harder datasets, such as CIFAR10 and CIFAR100 our method provides a more significant improvement in comparison to SGD.

For a more thorough comparison on the empirical performance of our proposed method we also run a continual learning task for text classification. We finetune BERT-base (Devlin et al., 2019) in the AG-News dataset (Gulli, 2005; Zhang et al., 2015) noticing still a minor improvement with our method in comparison to SGD.

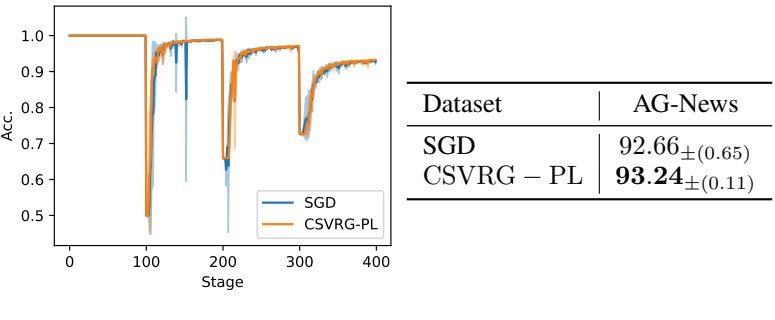

| (a) Continual learning evolution | (b) Final Acc. |
|---|---|

| Dataset | AG-News |
|---|---|
| SGD | $92.66_{\pm(0.65)}$ |
| CSVRG − PL | $\mathbf{93.24}_{\pm(0.11)}$ |

Figure 2: **Continual Learning in BERT+AG-News:** CSVRG − PL and SGD evolve similarly, with CSVRG − PL obtaining the best final performance at $93.24\%$ Acc.

### 4.3 UNLEARNING

In this section we evaluate our method on an unlearning task Sekhari et al. (2021); Guo et al. (2020). We initially start by training our model over the whole dataset for 30 epochs. Then starting from

one class we sequentially remove data of this class from the data set. For example in CIFAR10, we start by removing data of class 0 before continue with data from the other classes. Specifically we remove one batch of samples per stage and adjust the model with the remaining ones.

During the process of unlearning, we form two dynamic sets, the remember and the forget set. The forget set consists of all data that have been removed from the dataset, while the remember set consists of the ones remaining in it. The motivation behind unlearning is that the data might need to be removed from the trained model due to privacy reason or due to corruptions. The goal is that upon the arrival of a *remove request* to adjust the parameters of the model, so that the accuracy of the model on the forget set is as similar as possible to that of a model which has never seen these data, while maintaining high test accuracy on the remember set.

We measure the performance of the model at each stage, where a new batch of data is removed from the dataset. As we report on Table 3 the test accuracy for the remember set of both methods is comparable across all datasets, however, the accuracy on the forget set of SGD is significantly higher than that of our method, implying that $\mathrm{StochasticGradientDescent}$ (SGD) is unable to forget data, in contrast to $\mathrm{CSVRG - PL}$.

Table 3: We measure the final accuracy after 100 stages in the remember set (R Acc.) and the forget set (F Acc.). We compare the best performance for 200 steps, per stage, for MNIST, FashionMNIST, CIFAR10 and CIFAR100. $\mathrm{CSVRG - PL}$ consistently outperforms SGD by being less accurate in the forget set while minimally affecting the remember set performance.

| Dataset | MNIST | | FashionMNIST | | CIFAR10 | | CIFAR100 | |
| Metric | R Acc. ↑ | F Acc. ↓ | R Acc. ↑ | F Acc. ↓ | R Acc. ↑ | F Acc. ↓ | R Acc. ↑ | F Acc. ↓ |
|---|---|---|---|---|---|---|---|---|
| SGD | $99.18_{\pm(0.04)}$ | $95.17_{\pm(0.61)}$ | $\mathbf{94.57}_{\pm(0.11)}$ | $92.44_{\pm(0.72)}$ | $\mathbf{93.77}_{\pm(0.17)}$ | $87.61_{\pm(1.04)}$ | $\mathbf{76.43}_{\pm(0.23)}$ | $76.14_{\pm(0.51)}$ |
| $\mathrm{CSVRG - PL}$ | $\mathbf{99.19}_{\pm(0.02)}$ | $\mathbf{49.81}_{\pm(0.34)}$ | $94.48_{\pm(0.21)}$ | $\mathbf{71.67}_{\pm(3.01)}$ | $93.15_{\pm(0.18)}$ | $\mathbf{48.46}_{\pm(0.50)}$ | $73.50_{\pm(0.14)}$ | $\mathbf{44.81}_{\pm(0.34)}$ |

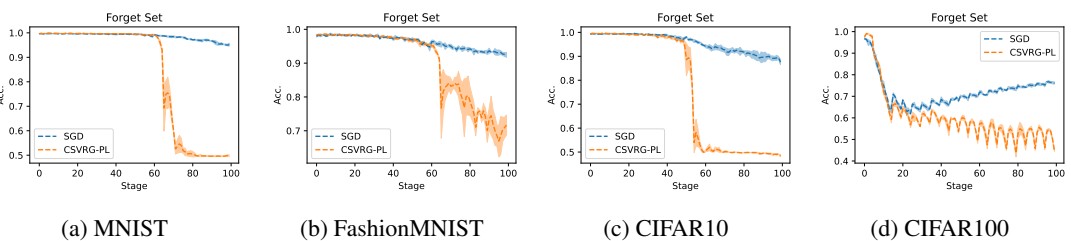

      (a) MNIST           (b) FashionMNIST          (c) CIFAR10          (d) CIFAR100

Figure 3: Accuracy of the model on the forget set.

## 5 CONCLUSION

In this work, we provide a first-order method $(\mathrm{CSVRG - PL})$ for Continual Finite Sum Minimization under the Polyak-Łojasiewicz condition. For accuracy $\epsilon > 0$, $\mathrm{CSVRG - PL}$ computes an $\epsilon$-approximate solution for all stages with $\mathcal{O}(n/\sqrt{\epsilon})$ FOs under the PL condition, for non-convex functions and recovers the rate of $\mathcal{O}(n/\epsilon^{1/3})$ of CSVRG for the strongly convex case with improved computation, simultaneously improving over the $\mathcal{O}(n/\epsilon)$ FO complexity of $\mathrm{StochasticGradientDescent}$ and over the $\mathcal{O}\left(n \log\left(1/\epsilon\right)\right)$ FO complexity of variance reduction methods. We also provide experimental evaluations indicating the effectiveness of our method in settings with time-evolving datasets like continual learning and unlearning.

**Limitations:** There is still an efficiency gap between the $\mathcal{O}(n/\sqrt{\epsilon})$ FO complexity of our method that applies to the non-convex setting and the $\Omega(n/\epsilon^{1/4})$ FO lower bound (Mavrothalassitis et al., 2024) developed in the convex setting. Closing this gap is a very interesting future research direction.

**Broader Impact:** This paper provides an efficient first-order method for continual finite-sum minimization. Our work contributes on the theoretical foundations of optimization and machine learning. We do not expect any negative societal impact from this work.

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

## A    Discussion on Assumption 2

In this section we will provide some more formal intuition supporting Assumption 2, for the sake of exposition we restate it up next.

**Assumption 2.** *(Diminishing returns) There exists a constant $K > 0$ such that for any stages $i > j$,*

$$\max_{x_i^\star \in \mathcal{X}_i^\star, x_j^\star \in \mathcal{X}_j^\star} \left\| x_i^\star - x_j^\star \right\|^2 \leq \frac{i-j}{i} K.$$

Formally we can show that the following for a compact set $\mathcal{D}$.

**Lemma 5.** *For all $i \in [n]$ and $j \in [n-i]$, if the prefix functions $g_i, g_j$ satisfy the quadratic growth property, then for every minimizer $x_i^\star$ there exists an $x_{j,i}^\star$ that minimizes $g_j$ over the compact set $\mathcal{D}$ and satisfies the inequality:*

$$\left\| x_i^\star - x_{j,i}^\star \right\| \leq \frac{i-j}{i} K$$

*Proof.* We start from the quadratic growth property for the function $g_i$

$$
\begin{aligned}
\left\| x_i^\star - x_{j,i}^\star \right\|^2 \;\leq\;& \frac{2}{\mu} \left( g_i(x_{j,i}^\star) - g_i^\star \right) \\
=\;& \frac{2}{\mu} \left( \frac{j}{i}(g_j(x_{j,i}^\star) - g_j(x_i^\star)) + \frac{1}{i} \sum_{k=j+1}^{i} \left( f_k(x_{j,i}^\star) - f_k(x_i^\star) \right) \right) \\
\leq\;& \frac{1}{i} \sum_{k=j+1}^{i} \left( f_k(x_{j,i}^\star) - f_k(x_i^\star) \right) \\
\leq\;& \frac{i-j}{i} G \left\| x_i^\star - x_{j,i}^\star \right\|
\end{aligned}
$$

In the third line we use the optimality of $x_{j,i}^\star$, which implies that: $g_j(x_{j,i}^\star) - g_j(x_i^\star) < 0$. On the last line we use the compactness of the set which implies that the functions $g_i, g_j$ are Lipschitz, for a constant $G$. □

**Corollary 2.** *For all $i \in [n]$ and $j \in [n-i]$, if the prefix functions $g_i, g_j$ satisfy the quadratic growth property, then over the compact set $\mathcal{D}$, it holds that:*

$$\min_{x_i^\star \in \mathcal{X}_i^\star, x_j^\star \in \mathcal{X}_j^\star} \left\| x_i^\star - x_{j,i}^\star \right\| \leq \frac{i-j}{i} K$$

As an immediate corollary, we also get that if the functions have unique minimizers then it holds that:

**Corollary 3.** *For all $i \in [n]$ and $j \in [n-i]$, if the prefix functions $g_i, g_j$ satisfy the quadratic growth property, then their minimizers $x_i^\star, x_j^\star$ over a compact set $\mathcal{D}$ satisfy the inequality:*

$$\left\| x_i^\star - x_j^\star \right\| \leq \frac{i-j}{i} K$$

# B    PROOF OF LEMMA 1

**Lemma 1** (Unbiased). *Let $\nabla_i^t$ be the gradient estimator as defined in Step 5 of Algorithm 2. Then for all $t \in [T_i]$ and for all $i \in [n]$, it holds that: $\mathbb{E}\left[\nabla_i^t\right] = \nabla g_i(x_i^t)$.*

*Proof.* Let $\mathcal{F}_i^t$ denote the filtration until step $t \in [T_i]$ of stage $i \in [n]$ By the definition of $\nabla_i^t$ we have that:

$$
\begin{aligned}
\mathbb{E}\left[\nabla_i^t | \mathcal{F}_i^t\right] &= \mathbb{E}\left[\nabla f_{u_t}(x_i^t) - \nabla f_{u_t}(\hat{x}_{prev}) | \mathcal{F}_i^t\right] + \tilde{\nabla}_i \\
&= \sum_{k=1}^i Pr\left[u_t = k\right]\left(\nabla f_k(x_i^t) - \nabla f_k(\hat{x}_{prev})\right) + \tilde{\nabla}_i \\
&= \frac{1}{i}\sum_{k=1}^i \left(\nabla f_k(x_i^t) - \nabla f_k(\hat{x}_{prev})\right) + \tilde{\nabla}_i \\
&= \frac{1}{i}\sum_{k=1}^i \left(\nabla f_k(x_i^t)\right) - \tilde{\nabla}_i + \tilde{\nabla}_i
\end{aligned}
$$

This concludes the proof, in the last line we used the property that $\tilde{\nabla}_i = \frac{1}{i}\sum_{k=1}^i \nabla f_k(\hat{x}_{prev})$, which is formally established in lemma 6. $\qquad\square$

**Lemma 6.** *At Step 12 of Algorithm 1, it holds that $\tilde{\nabla}_i = \sum_{k=1}^i \nabla f_k(\hat{x}_{prev})/i$*

*Proof of Lemma 6.* We will inductively establish Lemma 6. Notice that after stage $i = 1$, Algorithm 1 sets $\tilde{\nabla}_1 = \nabla f_1(\hat{x}_1)$. Up next we show that in case the induction hypothesis holds for stage $i - 1$ then it must essentially hold for stage $i$. Up next we consider the following 2 cases:

1. $i - prev \geq \alpha i$ meaning that $prev$ is updated in this stage.

2. $i - prev < \alpha i$.

Let us point out that the two cases are mutually exclusive.

For the first case Algorithm 1 reaches Step 5 and thus $\tilde{\nabla}_i = \sum_{j=1}^i \nabla f_j(\hat{x}_{i-1})/i$. At the same time $prev$ is set to $i - 1$ meaning that $prev = i - 1$ and $\tilde{\nabla}_i = \frac{1}{i}\sum_{k=1}^i \nabla f_k(\hat{x}_{prev})$. At the end of stage $i$ the algorithm sets $prev = i$ and $\tilde{\nabla}_i = \frac{1}{i}\sum_{k=1}^i \nabla f_k(\hat{x}_i)$, so our inductive hypothesis still holds for the next stage.

For the second case, from the inductive hypothesis we have that:

$$
\tilde{\nabla}_{i-1} = \frac{1}{i-1}\sum_{k=1}^{i-1} \nabla f_k(\hat{x}_{prev})
$$

At stage $i$, at Step 9 of Algorithm 1, which is always reached, since in this case $i - prev \leq \alpha \cdot i$. We have that $\tilde{\nabla}_{i-1}$ satisfies the inductive hypothesis. As a result,

$$
\begin{aligned}
\tilde{\nabla}_i &= \left(1 - \frac{1}{i}\right)\tilde{\nabla}_{i-1} + \frac{1}{i}\nabla f_i(\hat{x}_{prev}) \\
&= \frac{i-1}{i}\frac{1}{i-1}\sum_{k=1}^{i-1} \nabla f_k(\hat{x}_{prev}) + \frac{1}{i}\nabla f_i(\hat{x}_{prev}) \\
&= \frac{1}{i}\sum_{k=1}^i \nabla f_k(\hat{x}_{prev})
\end{aligned}
$$

$\qquad\square$

We note that the proofs for this section are similar to those of Mavrothalassitis et al. (2024).

## C    PROOF OF LEMMA 2

**Lemma 2** (Variance). *For the gradient estimator used in line* 5 *of Algorithm* 2 *we have:*

$$
\begin{aligned}
\mathbb{E}\left[\left\|\nabla_i^t - \nabla g_i(x_i^t)\right\|^2\right] &\leq 4\frac{L^2}{\mu}\mathbb{E}\left[g_i(x_i^t) - g_i(x_i^\star)\right] \\
&+ 4L^2\left(\frac{2}{\mu}\mathbb{E}\left[(g_{prev}(\hat{x}_{prev}) - g_{prev}(x_{prev}^\star))\right] + \frac{i - prev}{i}K\right)
\end{aligned}
$$

*where $K$ is the constant of Assumption 2.*

*Proof.* Let $\mathcal{F}_i^t$ denote the natural filtration with respect to iteration $t \in [T_i]$ of stage $i \in [n]$. By substitution from the definition of $\nabla_i^t$, we get:

$$
\begin{aligned}
\mathbb{E}\left[\left\|\nabla_i^t - \nabla g_i(x_i^t)\right\|^2 | \mathcal{F}_i^t\right] &= \mathbb{E}\left[\left\|\nabla f_{u_t}(x_i^t) - \nabla f_{u_t}(\hat{x}_{prev}) + \tilde{\nabla} - \nabla g_i(x_i^t)\right\|^2 | \mathcal{F}_i^t\right] \\
&= \mathbb{E}\left[\|\nabla f_{u_t}(x_i^t) - \nabla f_{u_t}(x_i^\star) - \nabla f_{u_t}(\hat{x}_{prev}) + \nabla f_{u_t}(x_i^\star)\right. \\
&+ \left.\tilde{\nabla} - \nabla g_i(x_i^\star) - \nabla g_i(x_i^t) + \nabla g_i(x_i^\star)\|^2 | \mathcal{F}_i^t\right] \\
&\leq 2\mathbb{E}\left[\left\|\nabla f_{u_t}(x_i^t) - \nabla f_{u_t}(x_i^\star) - \nabla g_i(x_i^t) + \nabla g_i(x_i^\star)\right\|^2 | \mathcal{F}_i^t\right] \\
&+ 2\mathbb{E}\left[\left\|\nabla f_{u_t}(\hat{x}_{prev}) - \nabla f_{u_t}(x_i^\star) - \tilde{\nabla} + \nabla g_i(x_i^\star)\right\|^2 | \mathcal{F}_i^t\right] \\
&= 2\mathbb{E}\left[\left\|\nabla f_{u_t}(x_i^t) - \nabla f_{u_t}(x_i^\star) - \nabla g_i(x_i^t) + \nabla g_i(x_i^\star)\right\|^2 | \mathcal{F}_i^t\right] \\
&+ 2\mathbb{E}\left[\|\nabla f_{u_t}(\hat{x}_{prev}) - \nabla f_{u_t}(x_i^\star) - \nabla g_i(\hat{x}_{prev}) + \nabla g_i(x_i^\star)\|^2 | \mathcal{F}_i^t\right] \\
&\leq 2\mathbb{E}\left[\left\|\nabla f_{u_t}(x_i^t) - \nabla f_{u_t}(x_i^\star)\right\|^2 | \mathcal{F}_i^t\right] \\
&+ 2\mathbb{E}\left[\left\|\nabla f_{u_t}(\hat{x}_{prev}) - \nabla f_{u_t}(x_i^\star)\right\|^2 | \mathcal{F}_i^t\right] \\
&\leq 2L^2\mathbb{E}\left[\left\|x_i^t - x_i^\star\right\|^2 | \mathcal{F}_i^t\right] \\
&+ 2L^2\mathbb{E}\left[\left\|\hat{x}_{prev} - x_i^\star\right\|^2 | \mathcal{F}_i^t\right] \\
&\leq 4\frac{L^2}{\mu}\mathbb{E}\left[g_i(x_i^t) - g_i(x_i^\star) | \mathcal{F}_i^t\right] \\
&+ 4L^2\left(\mathbb{E}\left[\left\|\hat{x}_{prev} - x_{prev}^\star\right\|^2 | \mathcal{F}_i^t\right] + \left\|x_{prev}^\star - x_i^\star\right\|^2\right) \\
&\leq 4\frac{L^2}{\mu}\mathbb{E}\left[g_i(x_i^t) - g_i(x_i^\star) | \mathcal{F}_i^t\right] \\
&+ 4L^2\left(\left\|\hat{x}_{prev} - x_{prev}^\star\right\|^2 + \frac{i - \text{prev}}{i}K\right) \\
&\leq 4\frac{L^2}{\mu}\mathbb{E}\left[g_i(x_i^t) - g_i(x_i^\star) | \mathcal{F}_i^t\right] \\
&+ 4L^2\left(\frac{2}{\mu}(g_{\text{prev}}(\hat{x}_{\text{prev}}) - g_{\text{prev}}(x_{\text{prev}}^\star)) + \frac{i - \text{prev}}{i}K\right)
\end{aligned}
$$

The third to last inequality follows from the smoothness of the functions $f_i$. The next inequality, follows from the quadratic growth property. The last inequality, follows from the quadratic growth property again. Taking expectation on both sides completes the proof. $\qquad\square$

## D  PROOF OF LEMMA 7

**Lemma 7.** *Under the update rule of line 5 of Algorithm 2, when using a step size $\gamma_t$, that satisfies the inequality $\gamma_t \leq \frac{\mu^2}{2L(L^2+\mu^2)}$, we have that:*

$$
\begin{aligned}
\mathbb{E}\left[g_i(x_i^{t+1}) - g_i(x_i^\star)\right] &\leq (1-\mu\gamma_t)\mathbb{E}\left[g_i(x_i^t) - g_i(x_i^\star)\right] \\
&+ 4\frac{L^3\gamma_t^2}{\mu}\mathbb{E}\left[(g_{prev}(\hat{x}_{prev}) - g_{prev}(x_{prev}^\star))\right] + 2L^3K\gamma_t^2\frac{i-prev}{i}
\end{aligned}
$$

*Proof.* We start our analysis from the smoothness of the function:

$$
\begin{aligned}
g_i(x_i^{t+1}) &\leq g_i(x_i^t) + \left\langle x_i^{t+1} - x_i^t, \nabla g_i(x_i^t)\right\rangle + \frac{L}{2}\left\|x_i^{t+1} - x_i^t\right\|^2 \\
&= g_i(x_i^t) - \gamma_t\left\langle \nabla_i^t, \nabla g_i(x_i^t)\right\rangle + \frac{L\gamma_t^2}{2}\left\|\nabla_i^t\right\|^2
\end{aligned}
$$

Let us upper bound the terms $-\gamma_t\left\langle \nabla_i^t, \nabla g_i(x_i^t)\right\rangle + \frac{L\gamma_t^2}{2}\left\|\nabla_i^t\right\|^2$:

$$
\begin{aligned}
&- \gamma_t\left\langle \nabla_i^t, \nabla g_i(x_i^t)\right\rangle + \frac{L\gamma_t^2}{2}\left\|\nabla_i^t\right\|^2 \\
&= (-\gamma_t + L\gamma_t^2)\left\langle \nabla_i^t, \nabla g_i(x_i^t)\right\rangle - L\gamma_t^2\left\langle \nabla_i^t, \nabla g_i(x_i^t)\right\rangle + \frac{L\gamma_t^2}{2}\left\|\nabla_i^t\right\|^2 \\
&= (-\gamma_t + L\gamma_t^2)\left\langle \nabla_i^t, \nabla g_i(x_i^t)\right\rangle - L\gamma_t^2\left\langle \nabla_i^t, \nabla g_i(x_i^t) - \nabla_i^t\right\rangle - \frac{L}{2}\gamma_t^2\left\|\nabla_i^t\right\|^2 \\
&\leq (-\gamma_t + L\gamma_t^2)\left\langle \nabla_i^t, \nabla g_i(x_i^t)\right\rangle + \frac{L\gamma_t^2}{2}\left\|\nabla g_i(x_i^t) - \nabla_i^t\right\|^2
\end{aligned}
$$

In the last line we use the identity:

$$
-\langle a, b\rangle - \frac{1}{2}\|a\|^2 \leq \frac{1}{2}\|b\|^2
$$

We set $D_i^t = g_i(x_i^t) - g_i(x_i^\star)$ and by substituting to the previous inequality, we get that:

$$
D_i^{t+1} \leq D_i^t - (\gamma_t - L\gamma_t^2)\left\langle \nabla_i^t, \nabla g_i(x_i^t)\right\rangle + \frac{L\gamma_t^2}{2}\left\|\nabla g_i(x_i^t) - \nabla_i^t\right\|^2
$$

By taking expectation conditional to the filtration up to step t, $\mathcal{F}_i^t$, we have that:

$$
\begin{aligned}
\mathbb{E}\left[D_i^{t+1}|\mathcal{F}_i^t\right] &\leq D_i^t - (\gamma_t - L\gamma_t^2)\left\|\nabla g_i(x_i^t)\right\|^2 + \frac{L\gamma_t^2}{2}\mathbb{E}\left[\left\|\nabla g_i(x_i^t) - \nabla_i^t\right\|^2|\mathcal{F}_i^t\right] \\
&\leq D_i^t - 2\mu(\gamma_t - L\gamma_t^2)D_i^t + \frac{L\gamma_t^2}{2}\mathbb{E}\left[\left\|\nabla g_i(x_i^t) - \nabla_i^t\right\|^2|\mathcal{F}_i^t\right]
\end{aligned}
$$

In the first line we used the unbiased property of our gradient estimator $\mathbb{E}[\nabla_i^t|\mathcal{F}_i^t] = \nabla g_i(x_i^t)$ as shown in Lemma 1. In the second inequality we used the PL-condition

$$
2\mu D_i^t \leq \left\|\nabla g_i(x_i^t)\right\|^2
$$

Note that the inequality is valid since $\gamma_t \leq \frac{\mu^2}{2L(L^2+\mu^2)}$ and we require $\gamma_t \leq \frac{1}{L}$, for $\gamma_t - L\gamma_t^2 \geq 0$.

For compactness for the rest of the proof, we denote as $B_i^t = \mathbb{E}[D_i^t] = \mathbb{E}[g_i(x_i^t) - g_i(x_i^\star)]$. By taking total expectation, over both sides of the previous inequality, we have:

$$
B_i^{t+1} \leq (1 - 2\mu\gamma_t + 2\mu L\gamma_t^2)B_i^t + \frac{L\gamma_t^2}{2}\mathbb{E}\left[\left\|\nabla g_i(x_i^t) - \nabla_i^t\right\|^2\right]
$$

Using Lemma 2 and substituting the upperbound for $\mathbb{E}\left[\left\|\nabla g_i(x_i^t) - \nabla_i^t\right\|^2\right]$, we get:

$$
\begin{aligned}
B_i^{t+1} &\leq (1 - 2\mu\gamma_t + 2\mu L\gamma_t^2)B_i^t \\
&+ \frac{L\gamma_t^2}{2}\left(4\frac{L^2}{\mu}B_i^t + 4L^2\left(\frac{2}{\mu}\mathbb{E}\left[(g_{prev}(\hat{x}_{prev}) - g_{prev}(x_{prev}^\star))\right] + \frac{i-prev}{i}K\right)\right) \\
&= (1 - 2\mu\gamma_t + 2\mu L\gamma_t^2 + 2\frac{L^3\gamma_t^2}{\mu})B_i^t \\
&+ 4\frac{L^3\gamma_t^2}{\mu}\mathbb{E}\left[(g_{prev}(\hat{x}_{prev}) - g_{prev}(x_{prev}^\star))\right] + 2L^3K\gamma_t^2\frac{i-prev}{i}
\end{aligned}
$$

By our assumption, that $\gamma_t \leq \frac{\mu^2}{2L(L^2+\mu^2)}$, we get that

$$(1 - 2\mu\gamma_t + 2\mu L\gamma_t^2 + 2\frac{L^3\gamma_t^2}{\mu}) \leq 1 - \mu\gamma_t$$

Substituting, gives the lemma statement and completes the proof. $\square$

## E   PROOF OF LEMMA 4

**Lemma 4.** *Let a sequence of functions $f_1, f_2, \ldots, f_i$ such that Assumption 2 is satisfied and the functions $g_i, g_j$ satisfy the quadratic growth property. Then,*

$$g_i(\hat{x}_j) - g_i^\star \leq \frac{i-j}{i}\frac{L \cdot K}{2} + C$$

*where $C = \frac{i-j}{i}L \cdot D \cdot \sqrt{\frac{2}{\mu}\left(g_j(\hat{x}_j) - g_j^\star\right)} + \frac{L}{\mu}\left(g_j(\hat{x}_j) - g_j^\star\right)$ and $K$ is the constant in Assumption 2.*

*Proof.*

$$
\begin{aligned}
g_i(\hat{x}_j) - g_i^\star &= g_i(\hat{x}_j) - g_i(x_j^\star) + g_i(x_j^\star) - g_i^\star \\
&\leq g_i(\hat{x}_j) - g_i(x_j^\star) + \frac{L}{2}\left\|x_j^\star - x_i^\star\right\|^2 \\
&\leq g_i(\hat{x}_j) - g_i(x_j^\star) + \frac{L}{2}\frac{i-j}{i}K
\end{aligned}
$$

In line 2 we used the smoothness of the function $g_i$. In line 3 we used assumption 2. Note that we select $x_j^\star$ above is the projection of $\hat{x}_j$ on the set $\mathcal{X}_j^\star$. So in order to get Lemma 4 it suffices to bound the first term of the right hand side $g_i(\hat{x}_j) - g_i(x_j^\star)$.

$$
\begin{aligned}
g_i(\hat{x}_j) - g_i(x_j^\star) &= \frac{1}{i}\sum_{k=1}^{i}\left(f_k(\hat{x}_j) - f_k(x_j^\star)\right) \\
&= \frac{1}{i}\sum_{k=j+1}^{i}\left(f_k(\hat{x}_j) - f_k(x_j^\star)\right) + \frac{j}{i}\sum_{k=1}^{j}\left(f_k(\hat{x}_j) - f_k(x_j^\star)\right) \\
&= \frac{1}{i}\sum_{k=j+1}^{i}\left(f_k(\hat{x}_j) - f_k(x_j^\star)\right) + \frac{j}{i}\left(g_j(\hat{x}_j) - g_j^\star\right)
\end{aligned}
$$

We will now focus on the terms $f_k(\hat{x}_j) - f_k(x_j^\star)$ for any $k$, we will start from the descent inequality, where we used the smoothness of the function $f_k$

$$
\begin{aligned}
f_k(\hat{x}_j) - f_k(x_j^\star) &\leq \langle \hat{x}_j - x_j^\star, \nabla f_k(x_j^\star)\rangle + \frac{L}{2}\left\|\hat{x}_j - x_j^\star\right\|^2 \\
&\leq \left\|\hat{x}_j - x_j^\star\right\|\left\|\nabla f_k(x_j^\star)\right\| + \frac{L}{2}\left\|\hat{x}_j - x_j^\star\right\|^2 \\
&\leq \left\|\hat{x}_j - x_j^\star\right\| \cdot L \cdot D + \frac{L}{2}\left\|\hat{x}_j - x_j^\star\right\|^2 \\
&\leq L \cdot D \cdot \sqrt{\frac{2}{\mu}\left(g_j(\hat{x}_j) - g_j(x_j^\star)\right)} + \frac{L}{\mu}\left(g_j(\hat{x}_j) - g_j(x_j^\star)\right)
\end{aligned}
$$

Where in the second inequality we used the Cauchy Schwartz inequality, in the third inequality, we used Assumption 1 and in the last one we used the fact that the function $g_j$ satisfies the quadratic growth property for all $j$. Substituting this upper bound in our equation we get:

$$
\begin{aligned}
g_i(\hat{x}_j) - g_i(x_j^\star) &\leq \frac{i-j}{i}\left(L \cdot D \cdot \sqrt{\frac{2}{\mu}\left(g_j(\hat{x}_j) - g_j(x_j^\star)\right)} + \frac{L}{\mu}\left(g_j(\hat{x}_j) - g_j(x_j^\star)\right)\right) \\
&\quad + \frac{j}{i}\left(g_j(\hat{x}_j) - g_j^\star\right) \\
&\leq \frac{i-j}{i}L \cdot D \cdot \sqrt{\frac{2}{\mu}\left(g_j(\hat{x}_j) - g_j(x_j^\star)\right)} + \frac{L}{\mu}\left(g_j(\hat{x}_j) - g_j(x_j^\star)\right)
\end{aligned}
$$

where in the second inequality we used the fact that $L \geq \mu$. Substituting this upperbound to the original inequality for $g_i(\hat{x}_j) - g_i^\star$, yields the lemma statement. $\qquad \square$

## F  PROOF OF LEMMA 3

**Lemma 3** (Convergence). *Under the update rule of line 5 of Algorithm 2, when using the step size $\gamma_t = 2/(\mu(t+c))$, with $c = 4L(\mu^2 + L^2)/\mu^3$, we have that:*

$$
\begin{aligned}
\mathbb{E}\left[g_i(x_i^{T+1}) - g_i(x_i^\star)\right] &\leq (c-1)(c-2)\frac{\mathbb{E}\left[g_i(\hat{x}_{i-1})\right] - g_i(x_i^\star)}{T^2} + 2L^3 K \frac{i - prev}{iT} \\
&+ \frac{16L^3\left(\mathbb{E}\left[(g_{prev}(\hat{x}_{prev}) - g_{prev}(x_{prev}^\star))\right]\right)}{\mu^3 T}
\end{aligned}
$$

*Proof.* By our selection of $\gamma_t = 2/(\mu(t+c))$, with $c = 4L(\mu^2 + L^2)/\mu^3$, we get that $\gamma_t \leq \frac{\mu^2}{2L(\mu^2+L^2)}$, for all $t \geq 0$, this allows us to use Lemma 7, for any $t$. To simplify notation and allow for compactness in our proof, we will denote as $B_i^t = \mathbb{E}\left[g_i(x_i^t) - g_i(x_i^\star)\right]$. We will also denote as $S_i = 4\frac{L^3}{\mu}\mathbb{E}\left[(g_{prev}(\hat{x}_{prev}) - g_{prev}(x_{prev}^\star))\right] + 2L^3 K \frac{i-prev}{i}$, to facilitate compactness in our proofs.

$$
\mathbb{E}\left[g_i(x_i^{t+1}) - g_i(x_i^\star)\right] \leq (1 - \mu\gamma_t)\mathbb{E}\left[g_i(x_i^t) - g_i(x_i^\star)\right] + \gamma_t^2 S_i
$$

By multiplying both sides of the inequality of the Lemma by $(t+c)(t+c-1)$, we get:

$$
\begin{aligned}
(t+c)(t+c-1)B_i^{t+1} \\
\leq (t+c-1)(t+c-2)B_i^t + (t+c)(t+c-1)\gamma_t^2 S_i
\end{aligned}
$$

By substituting $\gamma_t^2 = \frac{4}{\mu^2(t+c)^2}$, we get:

$$
\begin{aligned}
(t+c)(t+c-1)B_i^{t+1} \\
\leq (t+c-1)(t+c-2)B_i^t + \frac{4}{\mu^2}\frac{t+c-1}{t+c}S_i
\end{aligned}
$$

By summing the inequality from $t = 0$ to $T$, we get that:

$$
\begin{aligned}
\sum_{t=0}^{T}(t+c)(t+c-1)B_i^{t+1} \\
\leq \sum_{t=0}^{T}(t+c-1)(t+c-2)B_i^t + \frac{4}{\mu^2}\sum_{t=0}^{T}\frac{t+c-1}{t+c}S_i
\end{aligned}
$$

Now before proceeding let us point out two things. By setting $k = t+1$, the left hand side of the inequality can be rewritten as:

$$
\sum_{k=1}^{T+1}(k+c-1)(k+c-2)B_i^k
$$

Secondly it holds that

$$
\frac{t+c-1}{t+c} \leq 1
$$

By substituting these in the previous inequality we get:

$$
\begin{aligned}
\sum_{t=1}^{T+1}(t+c-1)(t+c-2)B_i^t \\
\leq \sum_{t=0}^{T}(t+c-1)(t+c-2)B_i^t + \frac{4}{\mu^2}\sum_{t=0}^{T}S_i
\end{aligned}
$$

A telescopic sum yields the following inequality:

$$
\begin{aligned}
(T+c)(T+c-1)B_i^{T+1} &\leq (c-1)(c-2)B_i^0 + \frac{4}{\mu^2}TS_i \\
B_i^{T+1} &\leq (c-1)(c-2)\frac{\mathbb{E}\left[g_i(\hat{x}_{i-1})\right] - g_i(x_i^\star)}{(T+c)(T+c-1)} + \frac{4TS_i}{\mu^2(T+c)(T+c-1)}
\end{aligned}
$$

Finally since $c \geq 1$, we have that: $(T+c)(T+c-1) \geq T^2$, so:

$$
B_i^{T+1} \leq (c-1)(c-2)\frac{\mathbb{E}\left[g_i(\hat{x}_{i-1})\right] - g_i(x_i^\star)}{T^2} + \frac{4S_i}{\mu^2 T}
$$

From here we directly get the lemma statement. $\qquad\square$

## G PROOF OF THEOREM 2

**Theorem 2.** *Let Algorithm 1 with parameters $\alpha = \sqrt{\epsilon/(10L^3K)}$, $\gamma_t = 2/(\mu(t+c))$ and,*

$$
T_i = \max\left( \sqrt{\frac{5LK}{2i\epsilon}}, c \cdot \sqrt{\frac{5L \cdot D}{i\sqrt{\epsilon}}}\sqrt{\frac{2}{\mu}}, \frac{80L^3}{\mu^3}, \sqrt{\frac{10L^3K}{\epsilon}} \right)
$$

*where $c = 4L(\mu^2 + L^2)/\mu^3$. Then for each stage $i \in [n]$ we have that*

$$
\mathbb{E}\left[g_i(\hat{x}_i)\right] - g_i^\star \leq \epsilon
$$

*where $\hat{x}_i$ is the output of Algorithm 2, for stage $i$.*

*Proof.* Since in the first stage we have only one sample it is easy to see that the algorithm reduces to gradient descent. Therefore after $T = \frac{1}{\sqrt{\epsilon}}$, we have a solution $\hat{x}_1$, such that:

$$
\mathbb{E}\left[g_i(\hat{x}_1)\right] - g_1^\star \leq \epsilon
$$

Now for subsequent stages let us assume that for all previous stages we have found an epsilon optimal solution in expectation. More formally at stage $i$, we have the inductive hypothesis, that for $j < i$

$$
\mathbb{E}\left[g_i(\hat{x}_j)\right] - g_j^\star \leq \epsilon
$$

Given our assumption that at each stage the step size is given by the equation $\gamma_t = 2/(\mu(t+c))$, we can use Lemma 3, from which, we have, for $T_i$ iterations, that:

$$
\begin{aligned}
\mathbb{E}\left[g_i(x_i^{T_i+1}) - g_i(x_i^\star)\right] &\leq (c-1)(c-2)\frac{\mathbb{E}\left[g_i(\hat{x}_{i-1})\right] - g_i(x_i^\star)}{T_i^2} + 2L^3K\frac{i - \text{prev}}{iT_i} \\
&+ \frac{16L^3\mathbb{E}\left[(g_{\text{prev}}(\hat{x}_{\text{prev}}) - g_{\text{prev}}(x_{\text{prev}}^\star))\right]}{\mu^3 T_i}
\end{aligned}
$$

Given our inductive hypothesis we have that:

$$
\mathbb{E}\left[g_{\text{prev}}(\hat{x}_{\text{prev}})\right] - g_{\text{prev}}(x_{\text{prev}}^\star) \leq \epsilon
$$

From Lemma 4, we have that:

$$
\begin{aligned}
\mathbb{E}\left[g_i(\hat{x}_{i-1})\right] - g_i(x_i^\star) &\leq \frac{L \cdot K}{2i} + \frac{L \cdot D}{i}\sqrt{\frac{2}{\mu}}\mathbb{E}\left[\sqrt{g_j(\hat{x}_j) - g_j^\star}\right] + \frac{L}{\mu}\mathbb{E}\left[g_j(\hat{x}_j) - g_j^\star\right] \\
&\leq \frac{L \cdot K}{2i} + \frac{L \cdot D}{i}\sqrt{\frac{2}{\mu}}\sqrt{\mathbb{E}\left[g_j(\hat{x}_j) - g_j^\star\right]} + \frac{L}{\mu}\mathbb{E}\left[g_j(\hat{x}_j) - g_j^\star\right] \\
&\leq \frac{L \cdot K}{2i} + \frac{L \cdot D}{i}\sqrt{\frac{2}{\mu}}\sqrt{\epsilon} + \frac{L}{\mu}\epsilon
\end{aligned}
$$

From line 5 of Algorithm 1 we ensure that $i - \text{prev} \le \alpha i$, so we have that:

$$
\begin{aligned}
\mathbb{E}\left[g_i(x_i^{T_i+1}) - g_i(x_i^{\star})\right] &\le (c-1)(c-2)\left(\frac{L \cdot K}{2iT_i^2} + \frac{L \cdot D}{iT_i^2}\sqrt{\frac{2}{\mu}}\sqrt{\epsilon} + \frac{L}{\mu T_i^2}\epsilon\right) + 2L^3K\frac{\alpha i}{iT_i} \\
&\quad + \frac{16L^3\epsilon}{\mu^3 T_i}
\end{aligned}
$$

We can decompose the right hand side of the previous inequality into 5 terms and demand that each of them is less than $\epsilon/5$, allowing for their sum to be less than $\epsilon$. So we have the following inequalities. For the first term:

$$
c^2\frac{LK}{2iT_i^2} \le \epsilon/5 \Rightarrow c \cdot \sqrt{\frac{5LK}{2i\epsilon}} \le T_i
$$

For the second term:

$$
c^2\frac{L \cdot D}{iT_i^2}\sqrt{\frac{2}{\mu}}\sqrt{\epsilon} \le \epsilon/5 \Rightarrow c \cdot \sqrt{\frac{5L \cdot D}{i\sqrt{\epsilon}}\sqrt{\frac{2}{\mu}}} \le T_i
$$

For the third term:

$$
\frac{L}{\mu T_i^2}\epsilon \le \epsilon/5 \Rightarrow \sqrt{\frac{5L}{\mu}} \le T_i
$$

For the forth term:

$$
2L^3K\frac{\alpha i}{iT_i} \le \epsilon/5 \Rightarrow 10L^3K\frac{\alpha}{\epsilon} \le T_i
$$

Using our selection of $\alpha = \sqrt{\epsilon/(10L^3K)}$ we get:

$$
T_i \ge \sqrt{\frac{10L^3K}{\epsilon}}
$$

For the fifth term, we get:

$$
\frac{16L^3\epsilon}{\mu^3 T_i} \le \epsilon/5 \Rightarrow T_i \ge \frac{80L^3}{\mu^3}
$$

By Taking a max over the required values for $T_i$, we get that for

$$
T_i = \max\{c \cdot \sqrt{\frac{5LK}{2i\epsilon}}, c \cdot \sqrt{\frac{5L \cdot D}{i\sqrt{\epsilon}}\sqrt{\frac{2}{\mu}}}, \frac{80L^3}{\mu^3}, \sqrt{\frac{10L^3K}{\epsilon}}\}
$$

It holds that

$$
\mathbb{E}\left[g_i(\hat{x}_i)\right] - g_i^{\star} \le \epsilon
$$

, so our induction holds and we get the theorem statement. $\qquad \square$

## H  OVERALL COMPLEXITY

In this section we calculate the total FO complexity of our algorithm. Before doing so, let us first prove Corollary 1, which for the sake of exposition we restate up next.

**Corollary 1.** *Over a sequence of $n$ stages, Algorithm 1 requires $2\sum_{i=1}^{n} T_i + 2n\lceil \log n/\alpha \rceil$ FOs.*

*Proof.* Algorithm 2 requires 2 FOs (Step 5) and thus Algorithm 2 requires overall $2T_i$ FOs during stage $i \in [n]$. At Step 6 and 15 Algorithm 1 requires at most $n$ FOs and thus by Lemma 8 it overall requires $2n\lceil \log n/\alpha \rceil$ FOs. $\qquad \square$

**Theorem 3.** *Let Algorithm 1 with parameters* $\alpha = \sqrt{\epsilon/(10L^3K)}$, $\gamma_t = 2/(\mu(t+c))$ *and,*

$$T_i = \max\left(\sqrt{\frac{5LK}{2i\epsilon}}, c \cdot \sqrt{\frac{5L \cdot D}{i\sqrt{\epsilon}}}\sqrt{\frac{2}{\mu}}, \frac{80L^3}{\mu^3}, \sqrt{\frac{10L^3K}{\epsilon}}\right)$$

*where* $c = 4L(\mu^2 + L^2)/\mu^3$. *Then the overall FO complexity of Algorithm 1 (across all* $n$ *stages) equals* $\mathcal{O}\left(n \log n \sqrt{1/\epsilon}\right)$.

*Proof.* From corollary 1 we have that the FO complexity of the algorithm is given by $3\sum_{i=1}^{n} T_i + 2n\lceil \log n/\alpha \rceil$. By taking the parameters $\alpha = \sqrt{\epsilon/(10L^3K)}$ and

$$T_i = \max\{c \cdot \sqrt{\frac{5LK}{2i\epsilon}}, c \cdot \sqrt{\frac{5L \cdot D}{i\sqrt{\epsilon}}}\sqrt{\frac{2}{\mu}}, \frac{80L^3}{\mu^3}, \sqrt{\frac{10L^3K}{\epsilon}}\}$$

we have that $2n \log n/\alpha = 2n\sqrt{\epsilon/(10L^3K)} \log n$ and

$$\sum_{i=1}^{n} T_i = \max\{c \cdot \sqrt{\frac{5LKn}{2\epsilon}}, c \cdot \sqrt{\frac{n5L \cdot D}{\sqrt{\epsilon}}}\sqrt{\frac{2}{\mu}}, n\frac{80L^3}{\mu^3}, n\sqrt{\frac{10L^3K}{\epsilon}}\}$$

$\square$

### H.1 PROOF OF LEMMA 8

In this section we prove Lemma 8, for the sake of exposition we restate it up next. The proof and the theorem are originally stated in Mavrothalassitis et al. (2024).

**Lemma 8.** *Over a sequence of* $n$ *stages, Algorithm 1 reaches Step* 5 *and* 12, $\lceil \log n/\alpha \rceil$ *times.*

*Proof of Lemma 8.* Step 5 and 12 are only executed when the following inequality is satisfied:

$$i - \text{prev} \geq \alpha \cdot i \Rightarrow i \geq \frac{1}{1-\alpha} \cdot \text{prev} \tag{2}$$

Once Algorithm 1 reaches Step 5 and 12 it necessarily, reaches Step 13 where prev is updated to $i$. Let $z_0 = 1, z_1, \ldots, z_k, \ldots$ the sequence of stages where $z_k$ denotes the stage at which Algorithm 1 reached Step 5 and 12 for the $k$-th time. By Equation 2 we get that $z_{k+1} \geq \frac{1}{1-\alpha} \cdot z_k$ implying that

$$z_k \geq \left(\frac{1}{1-\alpha}\right)^k$$

Since $z_k \leq n$ we get that $k \leq \frac{\log n}{\log\left(\frac{1}{1-\alpha}\right)}$. Notice that $\log\left(\frac{1}{1-\alpha}\right) = -\log(1-\alpha) \geq 1 - (1-\alpha) = \alpha$ and thus $k \leq \frac{\log n}{\alpha}$. $\square$

## I EXPERIMENTS

All of the experiments presented in the main part and the appendix were done on a single machine, with NVIDIA A100 SXM4 40 GB GPU.

In this section we include ablation studies and additional experiments.

### I.1 LEARNING RATE AND SCHEDULER SELECTION

As mentioned in the main part of the paper, we consider two schedulers, for our experiments. The motivation behind the linear scheduler lies in the construction of our method, since in order to achieve the theoretical guarantees, we have to opt for a linearly decreasing step size. The constant scheduler is just standard.

Here we give an extensive table according to which the optimal hyper parameters for our experiments were selected for the algorithms. We explained the procedure in Section 4.1, but for the sake of exposition, we restate it, before giving the numerical valuations.

We perform a grid search to select the learning rate for $\mathrm{CSVRG-PL}$ and SGD in the constant and linear schedules, we leave out a subset of $5\%$ of the training data as a validation set. We run every method and learning rate value for 3 random seeds and report the average and the final accuracy on the validation set.

Table 4: **Learning rate and scheduler selection:** We highlight the best learning rate value for each method and scheduler in **bold**.

| | | MNIST | | | | | |
|---|---|---|---|---|---|---|---|
| Method | Scheduler | Learning rate | | | | | |
| | | 0.001 | 0.005 | 0.01 | 0.05 | 0.1 | 0.5 |
| SGD | Constant | $95.90 \pm 0.44$ | $98.24 \pm 0.17$ | $98.50 \pm 0.08$ | $\mathbf{98.66} \pm 0.19$ | $98.16 \pm 0.27$ | $95.78 \pm 0.78$ |
| | Linear | $93.73 \pm 0.11$ | $97.87 \pm 0.13$ | $98.37 \pm 0.17$ | $98.59 \pm 0.20$ | $\mathbf{98.66} \pm 0.09$ | $96.68 \pm 0.57$ |
| $\mathrm{CSVRG-PL}$ | Constant | $96.41 \pm 0.08$ | $98.52 \pm 0.07$ | $98.62 \pm 0.22$ | $98.64 \pm 0.39$ | $\mathbf{98.78} \pm 0.42$ | $97.88 \pm 0.20$ |
| | Linear | $93.89 \pm 0.25$ | $98.31 \pm 0.01$ | $98.40 \pm 0.37$ | $\mathbf{98.91} \pm 0.32$ | $98.74 \pm 0.34$ | $98.33 \pm 0.52$ |
| | | FashionMNIST | | | | | |
| Method | Scheduler | Learning rate | | | | | |
| | | 0.001 | 0.005 | 0.01 | 0.05 | 0.1 | 0.5 |
| SGD | Constant | $81.68 \pm 0.07$ | $88.19 \pm 0.40$ | $88.82 \pm 0.74$ | $\mathbf{89.93} \pm 0.22$ | $89.51 \pm 0.26$ | $85.91 \pm 0.61$ |
| | Linear | $78.54 \pm 0.28$ | $86.91 \pm 0.28$ | $88.74 \pm 0.19$ | $90.26 \pm 0.16$ | $\mathbf{90.35} \pm 0.62$ | $87.44 \pm 0.42$ |
| $\mathrm{CSVRG-PL}$ | Constant | $82.13 \pm 1.04$ | $89.22 \pm 0.20$ | $89.02 \pm 0.85$ | $\mathbf{89.34} \pm 1.55$ | $88.48 \pm 1.65$ | $80.29 \pm 9.36$ |
| | Linear | $78.42 \pm 0.14$ | $87.72 \pm 0.43$ | $89.29 \pm 0.90$ | $\mathbf{91.05} \pm 0.11$ | $90.64 \pm 0.73$ | $88.81 \pm 0.56$ |
| | | CIFAR10 | | | | | |
| Method | Scheduler | Learning rate | | | | | |
| | | 0.001 | 0.005 | 0.01 | 0.05 | 0.1 | 0.5 |
| SGD | Constant | $50.07 \pm 0.53$ | $70.93 \pm 0.71$ | $70.44 \pm 9.68$ | $\mathbf{79.33} \pm 1.03$ | $77.11 \pm 1.84$ | $65.76 \pm 10.09$ |
| | Linear | $45.08 \pm 0.78$ | $67.21 \pm 0.78$ | $75.09 \pm 0.66$ | $80.89 \pm 0.50$ | $\mathbf{80.95} \pm 0.39$ | $70.55 \pm 5.80$ |
| $\mathrm{CSVRG-PL}$ | Constant | $51.61 \pm 2.15$ | $75.81 \pm 0.48$ | $78.99 \pm 1.26$ | $\mathbf{80.44} \pm 4.47$ | $78.45 \pm 4.15$ | $77.01 \pm 3.67$ |
| | Linear | $46.11 \pm 0.94$ | $69.48 \pm 0.96$ | $78.67 \pm 0.60$ | $85.80 \pm 0.32$ | $\mathbf{85.89} \pm 0.74$ | $81.51 \pm 4.45$ |
| | | AG-News | | | | | |
| Method | Scheduler | Learning rate | | | | | |
| | | 0.001 | 0.005 | 0.01 | 0.05 | 0.1 | 0.5 |
| SGD | Constant | $86.09 \pm 1.19$ | $90.47 \pm 0.26$ | $\mathbf{91.12} \pm 0.33$ | $25.00 \pm 0.00$ | $25.00 \pm 0.00$ | $25.00 \pm 0.00$ |
| | Linear | $82.61 \pm 3.41$ | $89.51 \pm 0.17$ | $\mathbf{90.74} \pm 0.10$ | $25.00 \pm 0.00$ | $25.00 \pm 0.00$ | $25.00 \pm 0.00$ |
| $\mathrm{CSVRG-PL}$ | Constant | $85.72 \pm 1.25$ | $90.29 \pm 0.04$ | $\mathbf{90.89} \pm 0.19$ | $25.00 \pm 0.00$ | $25.00 \pm 0.00$ | $25.00 \pm 0.00$ |
| | Linear | $79.84 \pm 2.80$ | $89.54 \pm 0.16$ | $\mathbf{90.54} \pm 0.21$ | $25.00 \pm 0.00$ | $25.00 \pm 0.00$ | $25.00 \pm 0.00$ |

## I.2 CONTINUAL LEARNING

In this experiment, we consider as many stages as batches in each dataset, i.e., $n = 600$ for MNIST/FashionMNIST and $n = 500$ for CIFAR10/100. Batches are sorted according to their label so that a new class label is introduced every $n/c$ stages, where $c$ is the number of classes. In order to be faithful to the continual learning setup, we select $T_i = 10$ in Algorithm 1. Setting $T_i$ to be larger, would result in re-training the model from scratch every time a new batch arrives, which we want to avoid. We select the best learning rate value and scheduler for each method according to Section 4.1.

We report the test accuracy in the classes we have visited during training. For example, in CIFAR10, in stages 0 to 49 we report the test accuracy for the samples with label "0" and in stages 450 to 499 the accuracy in the complete test set.

## I.3 UNLEARNING

In this section, we provide an extensive comparison for our Continual Unlearning Benchmark. In this setting, we seek to continuously remove samples from the dataset and measure the performance of the algorithm, in both the distribution of the data that has been removed, as well as the distribution of the data that have been retained in the training set. In order to do this we simply remove the deleted

samples from the training dataset of the algorithms, for both methods $CSVRG - PL$ and SGD. We repeat the procedure of removing batches of data for 100 stages and measure the generalization performance on the remember set and the forget set, with higher accuracy being more desirable for the former and less accuracy being preferable for the latter.

Table 5: **Forgetting tasks:** We measure the final accuracy after 100 stages in the remember set (R Acc.) and the forget set (F Acc.). We compare the performance when considering 10, 50 and 100 steps. $CSVRG - PL$ consistently outperforms SGD by being less accurate in the forget set while minimally affecting the remember set performance.

**MNIST**

| $T_i$ | 10 | | 50 | | 100 | | 200 | |
|---|---|---|---|---|---|---|---|---|
| Metric | R Acc. ↑ | F Acc. ↓ | R Acc. ↑ | F Acc. ↓ | R Acc. ↑ | F Acc. ↓ | R Acc. ↑ | F Acc. ↓ |
| SGD | $99.19_{\pm(0.04)}$ | $\mathbf{99.15}_{\pm(0.11)}$ | $\mathbf{99.25}_{\pm(0.02)}$ | $96.51_{\pm(0.27)}$ | $\mathbf{99.26}_{\pm(0.05)}$ | $95.83_{\pm(0.20)}$ | $99.18_{\pm(0.04)}$ | $95.17_{\pm(0.61)}$ |
| $CSVRG - PL$ | $\mathbf{99.26}_{\pm(0.03)}$ | $99.30_{\pm(0.04)}$ | $99.23_{\pm(0.04)}$ | $\mathbf{88.27}_{\pm(1.29)}$ | $\mathbf{99.26}_{\pm(0.02)}$ | $\mathbf{54.86}_{\pm(3.68)}$ | $\mathbf{99.19}_{\pm(0.02)}$ | $\mathbf{49.81}_{\pm(0.34)}$ |

**FashionMNIST**

| $T_i$ | 10 | | 50 | | 100 | | 200 | |
|---|---|---|---|---|---|---|---|---|
| Metric | R Acc. ↑ | F Acc. ↓ | R Acc. ↑ | F Acc. ↓ | R Acc. ↑ | F Acc. ↓ | R Acc. ↑ | F Acc. ↓ |
| SGD | $94.52_{\pm(0.06)}$ | $\mathbf{95.17}_{\pm(0.38)}$ | $94.75_{\pm(0.05)}$ | $93.18_{\pm(0.36)}$ | $94.61_{\pm(0.09)}$ | $92.82_{\pm(0.73)}$ | $\mathbf{94.57}_{\pm(0.11)}$ | $92.44_{\pm(0.72)}$ |
| $CSVRG - PL$ | $\mathbf{94.94}_{\pm(0.08)}$ | $95.27_{\pm(0.15)}$ | $\mathbf{94.87}_{\pm(0.06)}$ | $\mathbf{90.03}_{\pm(0.52)}$ | $\mathbf{94.70}_{\pm(0.11)}$ | $\mathbf{84.50}_{\pm(1.18)}$ | $94.48_{\pm(0.21)}$ | $\mathbf{71.67}_{\pm(3.01)}$ |

**CIFAR10**

| $T_i$ | 10 | | 50 | | 100 | | 200 | |
|---|---|---|---|---|---|---|---|---|
| Metric | R Acc. ↑ | F Acc. ↓ | R Acc. ↑ | F Acc. ↓ | R Acc. ↑ | F Acc. ↓ | R Acc. ↑ | F Acc. ↓ |
| SGD | $93.41_{\pm(0.18)}$ | $95.30_{\pm(0.22)}$ | $\mathbf{93.73}_{\pm(0.24)}$ | $90.49_{\pm(0.94)}$ | $\mathbf{93.85}_{\pm(0.18)}$ | $90.05_{\pm(1.10)}$ | $\mathbf{93.77}_{\pm(0.17)}$ | $87.61_{\pm(1.04)}$ |
| $CSVRG - PL$ | $\mathbf{94.05}_{\pm(0.15)}$ | $\mathbf{93.18}_{\pm(0.36)}$ | $93.71_{\pm(0.17)}$ | $\mathbf{57.40}_{\pm(2.15)}$ | $93.43_{\pm(0.04)}$ | $\mathbf{49.08}_{\pm(0.31)}$ | $93.15_{\pm(0.18)}$ | $\mathbf{48.46}_{\pm(0.50)}$ |

**CIFAR100**

| $T_i$ | 10 | | 50 | | 100 | | 200 | |
|---|---|---|---|---|---|---|---|---|
| Metric | R Acc. ↑ | F Acc. ↓ | R Acc. ↑ | F Acc. ↓ | R Acc. ↑ | F Acc. ↓ | R Acc. ↑ | F Acc. ↓ |
| SGD | $76.05_{\pm(0.19)}$ | $\mathbf{81.47}_{\pm(0.71)}$ | $\mathbf{76.12}_{\pm(0.20)}$ | $\mathbf{75.35}_{\pm(0.30)}$ | $\mathbf{76.19}_{\pm(0.21)}$ | $75.12_{\pm(0.52)}$ | $\mathbf{76.43}_{\pm(0.23)}$ | $76.14_{\pm(0.51)}$ |
| $CSVRG - PL$ | $\mathbf{76.47}_{\pm(0.07)}$ | $86.30_{\pm(0.18)}$ | $75.68_{\pm(0.15)}$ | $77.24_{\pm(0.49)}$ | $74.23_{\pm(0.18)}$ | $\mathbf{67.70}_{\pm(0.61)}$ | $73.50_{\pm(0.14)}$ | $\mathbf{44.81}_{\pm(0.34)}$ |

## I.4 CONTINUAL LEARNING IN THE TEXT CLASSIFICATION TASK

In this section, we replicate the continual learning experiments in Section 4 for the text classification task. We finetune BERT-base (Devlin et al., 2019) in the AG-News dataset (Gulli, 2005; Zhang et al., 2015).

We select the learning rate and schedule by training in a sample of 25% of the training set and evaluating in the remaining 75%. The results in Table 4 suggest the best learning rate value is 0.01 for both SGD and $CSVRG - PL$. Regarding the scheduler, unlike for the image datasets, the constant learning rate scheduler provides the best performance.

In Fig. 5 we can observe that $CSVRG - PL$ and SGD evolve similarly with $CSVRG - PL$ obtaining the best final accuracy at 93.24% v.s. 92.66 for SGD.

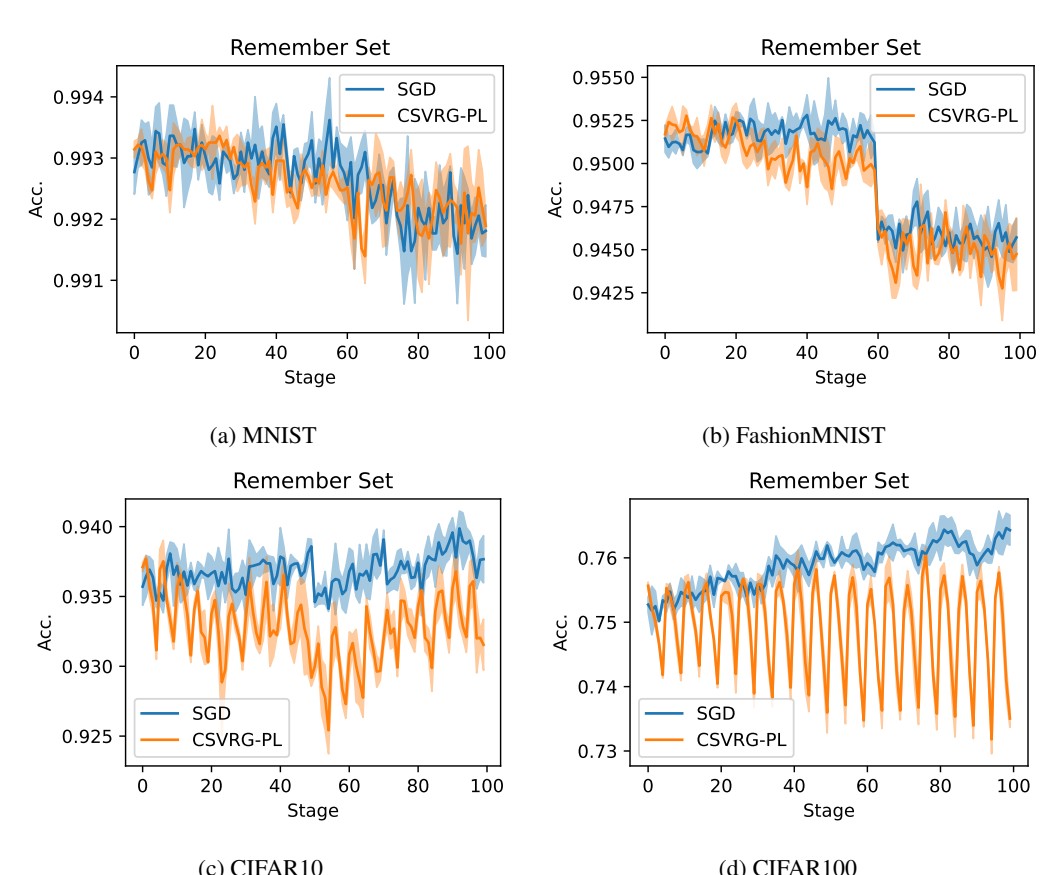

Figure 4: **Forgetting experiments:**In these plots we present the accuracy of the model on the remember set, when it is trained with SGD or SIOPT, for all stages.

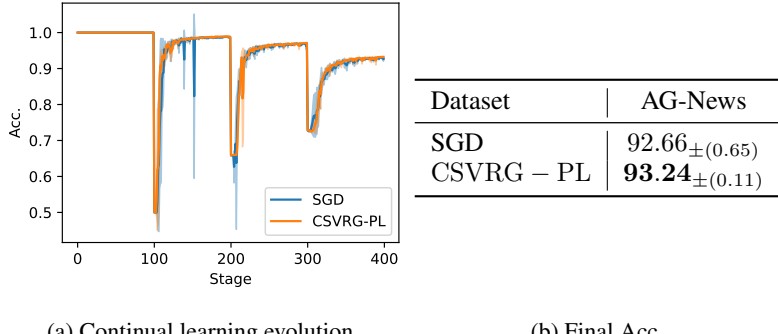

(a) Continual learning evolution

| Dataset | AG-News |
|---|---|
| SGD | $92.66_{\pm(0.65)}$ |
| CSVRG $-$ PL | $\mathbf{93.24}_{\pm(0.11)}$ |

(b) Final Acc.

Figure 5: **Continual Learning in BERT+AG-News:** CSVRG $-$ PL and SGD evolve similarly, with CSVRG $-$ PL obtaining the best final performance at $93.24\%$ Acc.

