# OpenReview forum: "CONTINUAL FINITE-SUM MINIMIZATION UNDER THE POLYAK-ŁOJASIEWICZ CONDITION"
_ICLR.cc/2025/Conference — Submitted to ICLR 2025_

### Official Review · Reviewer_Sm9X · 2024-11-03

**Soundness:** 3
**Presentation:** 2
**Contribution:** 2
**Rating:** 3
**Confidence:** 4

**Summary:**

The authors consider a continual version of finite-sum optimization (CFSM), where the loss functions arrive sequentially, and the ERM updates by including a newly received loss function in each iteration. This work extends the CFSM to the non-convex scenario by proposing a novel algorithm called CSVRG-PL. Specifically, when the loss functions in the sequence satisfy the PL property and a rather novel diminishing return assumption on the optimal points' distances, the authors demonstrate that CSVRG-PL achieves an improved rate of $O(n/\epsilon^{1/2})$, outperforming existing algorithms under the same setting.

**Strengths:**

Overall, the paper is well-structured and easy to follow. The contributions are clear, and the results offer an improvement over previous work under the CFSM framework and the PL condition. The comparison with existing methods is complete and allows readers to understand the background without effort.

**Weaknesses:**

However, a primary concern is the validity of the diminishing return assumption over distances between optimal points. This assumption is crucial to the improvement of the rate and appears unavoidable within the proof. Nevertheless, it is not justified in the previous literature including [Mavrothalassitis et al.,  2024], and may be too assertive. Specifically, in the non-convex setting, the optimal points of different loss functions could be arbitrarily distant from each other, making this diminishing return assumption quite unpractical. Besides, it seems that the authors want to defend this assumption by referencing the uniform stability condition in generalization theory. Indeed, uniform stability is often used as a powerful strategy for generalization error. However, it is typically a quantity that should be bounded by controlling the incremental error of gradient steps, rather than being assumed directly. As a result, this argument seems to be not supportive. It would be nice if the authors could provide more convincing arguments for this assumption.

**Questions:**

* It would be great if the authors could do more proofreading to promote the consistency of notations and referencing style.
* The discussion on the PL condition may be unnecessary, as it is a fairly standard assumption in non-convex optimization.
* In line 128, $O(n/\epsilon^{1/4})$ should be $\Omega(n/\epsilon^{1/4})$ because it is a lower bound result.

---

> ### Author Response · Authors · 2024-11-21
>
> We would like to thank the reviewer for taking the time to read through our manuscript.
>
> **Weaknesses**
>
> This assumption is intuitive, as it provides a mathematical framework for modeling diminishing returns in a cumulative sum of functions. Practically, this implies that the functions added to the sum contribute comparably. The reviewer contends that in non-convex settings, our assumption may not hold. However, our experiments, conducted under scenarios with slow distribution shifts done in the non-convex regime, suggest otherwise. Furthermore, the reviewer noted that this assumption is novel. However, prior work [1] establishes a stronger condition for the case of strongly convex functions, effectively implying the use of a more stringent version of Assumption 2. Additionally, in Appendix A, Corollary 3, we demonstrate that a stronger version of this assumption holds in a broader context beyond the strongly convex case.
>
> **Questions**
>
> We apologize if there is inconsistencies across the notation in the proofs. We would be happy to correct any such mistakes pointed out by the reviewer. We have corrected those made by Reviewer nLMq, Reviewer Sm9X as well as some additional ones through our own proof reading.

---

> ### Comment · Area_Chair_WASb · 2024-11-27
>
> Dear Reviewer,
>
> The authors have provided their rebuttal to your comments/questions. Given that we are not far from the end of author-reviewer discussions, it will be very helpful if you can take a look at their rebuttal and provide any further comments. Even if you do not have further comments, please also confirm that you have read the rebuttal. Thanks!
>
> Best wishes,
> AC

---

### Official Review · Reviewer_nLMq · 2024-11-04

**Soundness:** 3
**Presentation:** 3
**Contribution:** 1
**Rating:** 3
**Confidence:** 3

**Summary:**

This paper studies continual finite-sum minimization (CFSM) problems, introducing a novel first-order method, CSVRG-PL, and establishing its computational complexity under the PL condition. Numerical experiments are conducted to demonstrate the superiority of CSVRG-PL over SGD.

**Strengths:**

The paper is well-organized and easy to follow, with new theoretical contributions to the literature.

**Weaknesses:**

1. **Similarity to Prior Work**: The proposed algorithm is very similar to the method in [1], as both are inspired by the well-known SVRG method and follow the same algorithmic structure. As the authors point out, the primary difference lies in line 5 of Algorithm 2, with the rest of the method remaining nearly identical. This minor modification limits the novelty of the proposed approach.

2. **Limited Theoretical Contributions**: Although the paper presents three theorems, they essentially convey a single result—an upper bound for FO complexity under the PL condition. In contrast, [1] provides both upper and lower bounds, though it focuses only on the strongly convex setting and there remains a gap between the bounds. To enrich the content, it would be beneficial to explore a more general setting (e.g., the convex case without strong convexity or the non-convex case without the PL condition). Additionally, the authors could attempt to derive an improved lower bound or examine whether the gap between upper bounds in the strongly convex and PL settings is intrinsic to the problem structure. This would deepen the theoretical contribution of the paper.

[1] Ioannis Mavrothalassitis, Stratis Skoulakis, Leello Tadesse Dadi, and Volkan Cevher. Efficient continual finite-sum minimization. In The Twelfth International Conference on Learning Representations, 2024.

**Questions:**

1. In Corollary 1, the number of FO calls for lines 6 and 15 in Algorithm 1 is upper bounded by $n$, though the actual count is $i$. Could a finer analysis improve the result in Corollary 1?
2. In line 529, the authors mention achieving $O(n/\epsilon^{1/3})$ complexity for the strongly convex case. Could the authors clarify where this result appears?

[Typos]
Line 240: Remove the repeated "that."
Line 532: The complexity for variance reduction methods should be $O(n^2 \log (1/\epsilon))$.

---

> ### Author Response · Authors · 2024-11-21
>
> We would like to thank the reviewer for taking the time to read through our manuscript.
>
> **Weaknesses**
>
> 1. Regarding the similarity to prior work. We agree that the algorithm is similar to [1], however, apart from the difference of gradient estimator in line 5 of Algorithm 2, the output of Algorithm 2 is also different at line 9. While in the work of [1] the authors use average in order to recover their results, in our scheme we return the last iterate, which is a significant change by itself, for the analysis.
> 2. As discussed in our response to Reviewer xM2b, extending continual finite-sum optimization under the PL-conditions, require singificantly different proof techniques.
>
> **Questions**
>
> Regarding the questions:
> 1. We would like to thank the reviewer for their proposition, however, since lines 6 and 15 are only executed a logarithmic amount of times we speculate that this more careful analysis will not yield a significant improvement.
> 2. In our analysis we use Assumption 2. More specifically this assumption dictates how often we need to do a full gradient computation through the variance. In the strong convex case, the authors of [1] show a stronger inequality to this one. By substituting that inequality in our analysis it is possible to derive the rate of $O(n/\epsilon^{1/3})$

---

> > ### Comment · Reviewer_nLMq · 2024-11-26
> >
> > I thank the authors for their responses. I decide to maintain my score.

---

### Official Review · Reviewer_xM2b · 2024-11-05

**Soundness:** 2
**Presentation:** 3
**Contribution:** 2
**Rating:** 3
**Confidence:** 4

**Summary:**

The paper focuses on Continual Finite-Sum Minimization (CFSM) and proposes and analyzes a new algorithm, CSVRG-PL, for solving the problem when the functions satisfy the POLYAK-LOJASIEWICZ (a generalization of strong convexity that captures some structured non-convex problems as well). The paper includes experiments on deep learning problems.

**Strengths:**

The paper is well written, and under the strong assumption made in this work (see weaknesses below), the theoretical result presented in the paper looks correct. The experiments evaluate the proposed method CSVRG-PL in continual learning settings, which is not related to the assumptions made in the paper, but show the practicality of the proposed method.

**Weaknesses:**

### On restrictive setting:
The paper heavily relies on the paper of Mavrothalassitis et al., the first to propose algorithms in the Continual Finite-Sum Minimization for the strongly convex regime. The main contribution is the extensions of these ideas in the PL setting.

Extending strongly convex results to PL functions is a reasonable extension, but I do not consider this by itself to be enough of a contribution to justify a paper acceptance at NeurIPS. For this paper to make more sense, it would have been much more exciting and appealing if the theory was extended to convex problems or general non-convex (under potential growth conditions). From an optimization viewpoint, simply extending a strongly convex theory to PL is, most of the time, an easy task.

Despite the above comment, I appreciate the clearness in explaining the differences between the CFSM-PL and the work of Mavrothalassitis et al., as presented in point 3 of the main contributions, and I fully agree with this.
I understand the challenge of extensions to PL scenarios, but in my opinion, by comparing the two papers (the submitted one and Mavrothalassitis et al.), I do not see a difficulty in putting things together as this paper did.

### Issues with Theoretical assumptions

Let me highlight three two major flaws of the analysis:

1.  Assumption 1 is extremely strong, and even if it is used in previous/older papers, it does not mean that it should continue being used. I believe having it restricts the theorems substantially, and most likely, it does not hold for smooth PL functions in unconstrained setting (the D in the assumption can be extremely large for this to make sense in several practical examples, which will make the bounds on the theorem impractical)

2. Same holds for Assumption 2. In many practical scenarios without extra assumptions on interpolation (most likely the ones of the experiments in the paper as well), the K is extremely large and cannot really be quantified. The K appears in the upper bound of the theorems, which allows potentially very loose final results.

### On presentation and parts of the main paper

The whole paper has one single theorem (Theorem 2). I appreciate the detail on the theoretical justification of the paper, and I like the presentation of it, but in my opinion, all of this discussion can easily move to the appendix. I see a clear benefit in including all the lemmas in the main paper (considering the work of Mavrothalassitis et al., these lemmas are not difficult to prove to be worth part of the main paper). In addition, I am not sure why Theorems 1 and 3 are called Theorems (these are just a proposition and a corollary, respectively).
\end{itemize}

The experiments are not related to what the theory suggests. The papers directly solve DNNs, but this is not the theoretical focus of this work (PL functions). Not all ML papers should be about DNNs. Do the authors have any setting that their assumptions are actually satisfied and which is not toy example and not strongly convex?

**Questions:**

Please see my points and questions in the Weaknesses box

---

> ### Author Response · Authors · 2024-11-21
>
> We would like to thank the reviewer for taking the time to read through our manuscript.
>
> **Regarding the setting**
>
> We want to remark that extending variance reduction methods to the PL-condition is highly non-trivial, since circumventing the convexity assumption necessitates conceptually different techniques. Additionally we would like to request from the reviewer if he could be more specific about what kind of potential growth conditions  that he considers more interesting in the general non-convex case. For example the strong growth condition is much more difficult to satisfy than simple PL and in addition to that it allows for the acceleration of SGD to the deterministic rates, making the setting in general much less interesting.
>
> **Regarding the comments on the Assumptions**
>
> 1. We believe that the analogy with previous work creates some confusion here and we apologize. Similar assumptions in related work require that the gradient norm is bounded for any $x$, however, as clearly stated in our manuscript our assumption requires that the gradient norm of the functions is only bounded at $x^\star_j$, corresponding to the optimums of the finite sum, which is a much weaker and more reasonable assumption.
>
> 2. Assumption 2 is natural as well, since it just requires that the range of the optimums shrinks as we introduce more functions.
>
> We believe that our results provide the crucial first steps for extending the continual optimization framework beyong convexity.
>
> **Regarding the presentation on the main part**
>
> We included the main proof techniques in the main part so as to highlight the key theoretical challenges in extending continual finite-sum optimization for PL conditions.
>
> Regarding the experimental evaluations, there has been a line of work arguing that overparametrized neural networks have a landscape that satisfies the PL inequality; this fact as well as the general interest in DNNs made us opt for this experimental. We believe that presenting experimental evaluations, related to real-world applications, is a merit and not a weakness of our work.

---

> > ### Comment · Reviewer_xM2b · 2024-11-27
> >
> > Thanks for the response.
> >
> > With growth condition, I refer to the different relaxed growth conditions that appear in the literature (not the strong growth). For this check, for example, section 3 of [1]. The PL condition is an assumption of the problem, while the growth condition is used to bind the noise of the stochastic algorithm.
> >
> > The PL condition indeed, requires different proof techniques compared to a stongly convex regime. Still, I would not call them highly non-trivial (several papers have already proposed variance-reduced methods for PL functions).
> >
> > Again, I thank the authors for their responses. I decide to maintain my score.
> >
> > [1] Khaled, Ahmed, and Peter Richtárik. "Better theory for SGD in the nonconvex world." arXiv preprint arXiv:2002.03329 (2020).

---

> ### Comment · Area_Chair_WASb · 2024-11-27
>
> Dear Reviewer,
>
> The authors have provided their rebuttal to your comments/questions. Given that we are not far from the end of author-reviewer discussions, it will be very helpful if you can take a look at their rebuttal and provide any further comments. Even if you do not have further comments, please also confirm that you have read the rebuttal. Thanks!
>
> Best wishes,
> AC

---

### Meta-Review · Area_Chair_WASb · 2024-12-05

**Metareview:**

This paper studied finite-sum minimization under the Polyak-Łojasiewicz condition (PL). New theoretical results using first-order methods were provided, and these results were also supported by experimental results. Although the reviewers found the paper well-organized and it has theoretical contributions, there were general concerns about the assumptions being too strong and the contributions being incremental over previous work, in particular the paper by Mavrothalassitis et al. in ICLR 2024. The scores are unianimously on the rejection side, and the decision for the meta-review is hence rejection.

**Additional Comments On Reviewer Discussion:**

The reviewers mainly raised concerns on restricted assumptions and the theoretical contribution being incremental. The authors briefly clarified on those points but did not convince the reviewers.

---

### Decision · Program_Chairs · 2025-01-22

Reject